

# On the path to the digital rock physics of gas hydrate bearing sediments – processing of in-situ synchrotron-tomography data

Kathleen Sell[1], Erik-H. Saenger[2], Andrzej Falenty[3], Marwen Chaouachi[3], David Haberthür[4], Frieder Enzmann[1], Werner F. Kuhs[3] & Michael Kersten[1]

[1]Institute of Geosciences, Johannes Gutenberg-University, Mainz, Germany
[2]International Geothermal Centre & Ruhr University, Bochum, Germany
[3]GZG Crystallography, Georg-August-University, Göttingen, Germany
[4]Swiss Light Source, Paul Scherrer Institute, Villigen, Switzerland

*Correspondence to:* Kathleen Sell (sell@uni-mainz.de)

**Abstract.** To date, very little is known about the distribution of gas hydrates in sedimentary matrices and the resulting matrix-pore network affecting the seismic properties at low hydrate concentration. Digital rock physics offers a unique solution to this issue yet requires good quality, high resolution 3D representations for the accurate modelling of petrophysical and transport properties. Although such models are readily available *via* in-situ synchrotron radiation X-ray tomography the analysis of such data asks for complex workflows and high computational power to maintain valuable results. Here, we present a best-practise procedure complementing data from Chaouachi et al. (*Geochemistry, Geophysics, Geosystems* 2015**,** *16* (6), 1711-1722) with data post-processing, including image enhancement and segmentation as well as numerical simulations in 3D using the derived results as a direct model input. The method presented opens a path to a model-free deduction of the properties of gas hydrate bearing sediments when aiming for in-situ experiments linked to synchrotron-based tomography and 3D modelling.

*Keywords:* synchrotron-based tomography, image-enhancement, segmentation, gas-hydrates, digital rock physics



## Introduction

With the continuous improvement and popularization of high resolution visualization methods (Holzer and Cantoni, 2012; Desbois et al., 2013; Hemes et al., 2015; Cnudde and Boone, 2013; Liu et al., 2016; Deville et al., 2013; Berg et al., 2013; Pak et al., 2015) digital rock physics gained a firm foothold in the geophysical world as a potent toolset in studies of rock properties

(e.g. porosity and permeability) and pore-scale processes like e.g. electric and heat-conductivity or propagation of acoustic waves (Andrä et al., 2013b, a; Madonna et al., 2013; Saenger et al., 2011). In this approach, numerical simulations of transport and petrophysical properties are conducted on realistic rock representations that ultimately can be up-scaled and compared to remote sensing data. The advantage of this method over traditional and often oversimplified models lays in a more faithful description of complex pore geometries and microstructures found in natural formations (Andrä et al., 2013b, a). The most

common base for the numerical simulation are segmented 3D reconstructions from various nondestructive X-ray computed tomography (X-ray CT) techniques. With a good tradeoff between the level of detail and investigated sample volume these analytical tools became regularly used on geological samples at ambient or cryogenic conditions (Murshed et al., 2008; Klapp et al., 2012; Pak et al., 2015; Berg et al., 2013; Wang et al., 2016; Wang et al., 2015).

More recently digital rock physics took also on data from a fairly new group of techniques focused on *in-situ* studies recreating

complex settings that cannot be easily accessed with conventional means. One of such difficult environments are certainly marine and permafrost strata that are host to crystalline gas-water compounds known as gas hydrates (GH-s) (Sloan and Koh, 2008). Composed predominantly of methane, GHs are a target of intensive geophysical surveys, drilling and well-logging operations aiming on the identification and quantification of natural deposits of this unconventional source of hydrocarbons (Sloan and Koh, 2008; Boswell and Collett, 2011; Moridis et al., 2011). The identification of potential deposits with remote

sensing methods largely relays on high electric resistivity or characteristic seismic anomalies in which the increased velocity of seismic wave is coupled with high attenuations (Best et al., 2010; Guerin and Goldberg, 2005; Matsushima et al., 2015; Priest et al., 2006). The latter phenomenon has been earlier interpreted as losses due to squirt flow in interfacial liquids trapped between mineral frame and GH crystals (Priest et al., 2006) but became confirmed only in recent sub-μm CT studies (Chaouachi et al., 2015).

The quantification of GH saturation levels is not straightforward as there is very little known about the formation, microstructure and distribution of hydrates in natural settings, parameters fundamental to the interpretation of geophysical exploration. Different habits, distributions and saturation of gas hydrate crystals in the pore space affect the physical properties of the hydrate-bearing sediment (Priest et al., 2005; Waite et al., 2004). As the recovery of unperturbed natural methane hydrates is very difficult due to their fast decomposition under ambient conditions, a number of researchers have attempted to

recreate the natural environment of gas hydrate in sedimentary matrices *via* laboratory experiments (Berge et al., 1999; Best et al., 2010; Best et al., 2013; Dai et al., 2012; Dvorkin et al., 2003; Ecker et al., 2000; Hu et al., 2010; Li et al., 2011; Priest et al., 2006; Priest et al., 2009; Spangenberg and Kulenkampff, 2006; Yun et al., 2005; Zhang et al., 2011). This collective effort eventually leads to a set of idealized micro-structural models (Figure 1) but the approximations turned out to be still far



from being satisfactory. None of the simplified models could accurately predict GH saturations from field electric resistivity or seismic data alone (Waite et al., 2009; Dai et al., 2012). This might now change with the advent of high resolution *in-situ* X-ray CT methods that open new possibility to explore various nucleation and growth paths of GH in a sedimentary matrix in three dimensions with a pixel resolution below 1μm. The application of digital rock physics to such complex system is certainly

non-trivial due to a number of constraints imposed by the experimental setups, e.g. fixed sample-detector distance, high attenuation of the X-ray beam, low contrast, but still remains a viable way to improve the quantification of GH contents and to characterize the properties of the system.

The primary challenge is found in the acquisition of high quality 3D models that are capable of capturing fine features such as micro-cracks, fine porosity and grain-to-grain contacts that could potentially influence the computed properties (Kerkar et al.,

2014; Chaouachi et al., 2015). Another critical point specific for the seismic anomaly in GH sediments is a clear identification of the at most a few μm thick water film separating mineral frame and GHs. As we demonstrated in our recent work (Chaouachi et al., 2015; Falenty et al., 2015) this high requirements can be meet by synchrotron sources due to their unmatched resolution, high photon density and tunable energy of the beam. Yet, even in this case the semi-automatic segmentation and labeling is a difficult task due to a low density contrast and refractory edge enhancement. In this study, we present a workflow for the image

enhancement, segmentation and labelling of this data for further 3D modelling.

## 1 Data acquisition

In the following section the *in-situ* experiment and scanning procedure will be described briefly as details are given in (Chaouachi et al., 2015; Falenty et al., 2015). As the nucleation and formation of hydrates within the pore space occurs on the nano- to microscale, high-resolution synchrotron radiation X-ray tomography (SRXCT) was the best experimental choice.

**1.1 In-situ experiment**

SRXCT scans were obtained at the TOMCAT beamline of the Swiss Light Source (SLS), Paul Scherrer Institute (PSI) in Villigen, Switzerland (Stampanoni et al., 2006). The high brilliance, flexible selection of the energy window and exceptional resolution of this setup allows for studies of dynamic processes in complex environmental cells with an excellent signal-to-noise ratio and fast acquisition time (Figure 2A). The formation process was followed in a custom built in-situ cell composed

of: 1) an aluminium base, 2) sample holder mounted on top of the base and 3) TECAPEI (Polyetherimid) dome used to seal the setup (Figure 2B). Temperature control is provided by a Peltier element mounted with the cold side at the bottom of the aluminum base. The temperature of the sample is actively controlled via a PID controller with long term stability within ± 0.1°C. In order to avoid potential vibrations from a pump the hot side of the Peltier element is cooled with a laminar gravity-driven flow of water. The pressure inside the cell is measured with an ASHCROFT KXD pressure sensor and the data were

recorded every 5 s on a computer. The sample holder with a wall thickness of 0.1mm, an inner diameter of 2 mm and a length



of 10 mm is made also out of aluminum alloy which ensures a good thermal conductivity to lower the temperature T to a constant of T = 2 °C. The size of the sample holder was chosen to fulfil the size requirements of the numerical modelling planned thereafter but also to meet the limitations concerning beam energy attenuation (Figure 2B). Using methane gas would lead to considerable additional challenges because of high required gas pressures during the measurements and a more difficult

data processing due to a low density difference between water and methane hydrate (Jin et al., 2006). In order to mitigate these complications, in our experiments methane was substituted with xenon (Xe) gas, known to be equivalent with respect to the resulting gas hydrate in all important physical properties; xenon hydrate forms at very moderate p-T conditions of T=2°C and a very moderate pressure of p ≥ 0.2 MPa. The ambient atmosphere in the closed volume is replaced *via* several compression and decompression cycles with pure Xe gas while staying below the thermodynamic stability boundary of Xe-hydrate

(Chaouachi et al., 2015). In this study, the focus is on samples containing natural quartz sand of about 200 – 300 μm grain size. Details on this sedimentary material can be found in (Chuvilin et al., 2011).

## 1.2 Scanning procedure

For each tomogram, 3201 (or 1501) projections at an integration time of 150 ms (or 350 ms) each were acquired over a sample rotation of 180° with a monochromatic X-ray beam energy of 21.9 keV. Two different objectives (UPLAPO10× or

UPLAPO20×) with an optical 10-folded or 20-folded magnification were used, respectively. After penetrating the sample, X-rays were converted into visible light by a 20 μm thick scintillator LuAG:Ce and captured by a high-resolution CCD camera of 2048×2048 pixels. The data has been reconstructed out-of-cam by a gridded Fourier transform-based algorithm (Marone and Stampanoni, 2012). The reconstruction process yields in an image matrix of 2560 x 2560 x 2160 voxels, with an isometric voxel size of 0.74 μm and 0.38 μm at 10-folded and 20-folded magnification, respectively. Time-resolved scanning was

achieved by starting the reaction, stopping it after a prior lab-tested reaction time by reducing pressure to the stability boundary, with allowance for the system to stabilize, followed by subsequent scanning (Chaouachi et al., 2015; Falenty et al., 2015). This procedure is called "stop-and-go in situ tomography" which repeated multiple times to follow consecutive stages of hydrate formation.

## 2. Data Processing

**2.1 Reconstruction**

To translate the sinograms to 3D tomographic images the regridding reconstruction algorithm (Marone and Stampanoni, 2012;Stampanoni et al., 2006) was used. The reconstructed tomograms revealed that in spite of contrast enhancement with Xe gas the grey value differences between the gas and water phase were low. Moreover, a correct visualisation of the thin fluid layer located in-between the grains and the hydrate (Chaouachi et al., 2015) required a new approach to enhance the image



contrast. For this, the original projections of a scan were recreated to obtain the original sinograms. Following the refraction index of the Henke database with water as the starting value (Henke et al., 1993) for a beam energy of 21.9 keV the material's refractive index deviates from unity by the real part $\delta = 4.808 \times 10^{-7}$ and by the imaginary part $\beta = 2.563 \times 10^{-10}$. The Paganin algorithm (Paganin et al., 2002) is able to extract phase information in samples, which was tested on the present data to recover

contrast between the xenon and the aqueous phase. A representative sample has been chosen to apply the Paganin iteration. For this test, the sample to detector distance was set to 25 mm and the rotation centre to 1281.2. In total, 25 different parameters were chosen and applied to various 2D slices. The reconstruction for each set of parameters did not meet the expectations to show a large and clear border between Xenon and water (Figure 3).

## 2.2 Image Enhancement

At this point the reconstructed images are limited to the spatial resolution and affected by image artifacts but already enable to distinguish between gas, water, sediment grains and hydrate. Image artifacts that occurred most in the samples were inhomogeneity in grey values, streaks resulting from bad rotation alignment and edge enhancement.

The first step in data post-processing is to reduce image noise and scan artifacts using common image filters, calibrated for the appropriate dimensions and kernel window sizes (Sell et al., 2015). The software package *Avizo* 7.1 (FEI, France) has been

used for the steps of image filtering and segmentation. One must be aware that every step of image enhancement changes the original data set affecting subsequent steps required for data analysis (Sell et al., 2013b). Working on big datasets often stresses the computational power to an extent where an effective processing is not accomplishable anymore. Therefore it is highly recommended to split the datasets into smaller sub-volumes before taking further action of data post-processing. It is essential to deploy the same enhancement steps and parameters for each digitally sampled sub-volume. *Avizo* offers several options

including the *Extract Subvolume function*, the *Split Volume function* and the *Region of Interest* cropping tool to gain smaller sub-volumes. The cropping tool is not recommended for the described situation as it substitutes the complete dataset by the selected region of interest instead of gaining several smaller volumes in one attempt. For this study, the *Extract Subvolume* function was the best choice. As the full datasets each have a known size of 28 GB equal to 2560 x 2560 x 2160 pixels it was decided to split the dataset to 8 sub-volumes each of a size of 1280 x 1280 x 1080. Subsequently the sub-volumes are rated in

terms of image artifacts by considering various slices in all xy-, xz- and yz-direction. Choosing the right image enhancement technique might require an extensive testing of the available filters prior further steps. Depending on the dataset and the characteristics of the image artifacts and noise, filters have different impact on the data. Therefore, as we operate at the limiting edge of gaining image quality and running into data bias, considerable caution must be exerted when treating this multi-phase system.



### 2.3 Preliminary Study on image enhancement

A preliminary study on the effect of various image filters on modelling result was necessary to determine the best option of filters aiming at high quality images to gain volume rendered images (Figure 4). For the pre-study well-known and established filters from former case studies (Sell et al., 2013a; Madonna et al., 2013; Shulakova et al., 2013) have been applied to a Region of Interest (400 x 400 x 400 voxels) cropped from the original raw dataset. Five datasets which derived from samples containing approximately 17 Vol% of hydrate at full-transformation state were selected. The knowledge on the hydrate bulk in the samples will be used as a proxy later. In the following we provide a list of the applied filters and settings used in this study.

**Anisotropic Diffusion Filter** (*AD*): The concept of the *AD* filter is to smooth out noise in predefined areas of an image, but stopping at sharp edges representing boundaries between phases. This way, edges and sharp boundaries between phases are preserved, and image noise is significantly reduced (Kaestner et al., 2008;Porter and Wildenschild, 2010). A comparison of the current voxel with the grey values of its 6 neighbors takes place and diffusion is fulfilled when the *threshold stop criterion* is not exceeded. If the difference between one voxel and its six adjacent neighbors exceeds the given value no diffusion takes place. Another option to control the diffusion process of the filter is to reduce, or increase the *diffusion time*. The parameter *number of iteration* defines how often the algorithm will be used on the data. The bigger this number is, the more blurred is the resulting image. Smoothing is performed by applying a Gaussian filter. For our investigations the *threshold stop criterion* was set to the value 22968 as this is the approximated transition of the grain phase to hydrate. *AD* was run on CPU device with 5 iterations. **Adaptive Histogram Equalization Filter** (*AHE*): The *AHE* filter performs a so-called contrast limited adaptive histogram equalization (CLAHE) on the data set. The CLAHE algorithm in *Avizo* partitions the images into contextual regions and applies the histogram equalization to each one. This evens out the distribution of used grey values and thus makes hidden features of the image more visible (Reza, 2004). For this study the *brightness* was set to 0.716, the *contrast* to 0.41 and the *clip limit* to 7 after trying various values ahead to gain a reasonable result. **Edge Detection Moments Filter** (*EDM*): The *EDM* is a statistical feature detection to mask out noisy regions or for edge detection. Here, the brightness was set to 0.5 and *contrast* was set to 2. **Edge Detection Sobel Filter** (*EDS*): The *EDS* filter is based on the Sobel operator which preserves grain boundaries by searching for the most homogeneous feature of the input data and assigns the averaged value to an elementary volume where it highlights boundaries between different materials (Shulakova et al., 2013). For this study, the *EDS* was set to a *brightness* of 0.639 at a *contrast* of 2. **Morphological Dilatation Filter** (*MD*) & **Morphological Erosion Filter** (*ME*): The *MD* noise reduction filter was applied to replace the value of a pixel by the largest value of neighboring pixels. The *shape* of the neighborhood can be defined via the input field *Neighborhood* which was set to 8. The *ME* filter works contrary to the *MD* by replacing the value of a pixel by the smallest value of 8 neighboring pixels. **3D Median Filter** (*M*) is a simple edge-preserving filter which was set to a kernel size of 3. This filter is calculating the mean grey value of neighboring voxels, in our case 9 since the filter was executed in 3D mode, subsequently the initial voxel is replaced by the resulting mean grey value. The **Unsharp Masking Filter** (*UM*) sharpens an image using an *unsharp mask*. In the *UM* approach for image enhancement, a weighted fraction of the highpass filtered version of the image which results in the *unsharp mask*, is added to the original



image (Ramponi, 1999). The *unsharp mask* is computed in *Avizo* by a Gaussian filter of a certain kernel size. For this study the *UM* was applied with different sharpness settings of 0.5 for UM1 and 1 for UM2 at a kernel size of 3. The **Non-local Means Filter** (*NLM*), implemented in *Avizo*, is a windowed version of the non-local means algorithm (Buades et al., 2005a;Buades et al., 2005b). The main aim is to denoise data based on comparing voxels for similarities in a selected window

in which a new weight for a voxel is assigned. After a Gauss kernel was run on the weighted values, the new value will be assigned replacing the former grey values. The filter is most efficient, if the image is affected by white noise. In *Avizo* the parameters *window size*, the *local neighborhood*, and the *similarity value* can be customized. The *NLM* filter is also an appropriate tool for *Salt and Pepper* denoising which is caused by image sensor defects (Sarker et al., 2012). In this study, the *NLM* filter was run in 3D mode on CPU device. The search window was equal to 21, the local neighborhood set to 6 at a

similarity value of 0.71.

Adjacent all filtered datasets were segmented using the same routine which will be described in the following section 'Segmentation'. For this, binary values of 0, 1 and 2 were assigned to the pore space, grains and hydrate phase, respectively. Three of the nine filtered volumes have not been binarized as the necessary segmentation routine was not applicable to result

in datasets intended for numerical investigation. The module PoroDict of the software package GeoDict (Math2Market) was executed on the binarized images to quantify the hydrate in the filtered datasets. Even though the effects on image quality may not be obvious at first glance, further numerical quantification of the hydrate phase revealed significant differences (Figure 5).

In Figure 6 all results regarding the image filter run on 5 representative volumes are depicted. According to the numerical

results of the described preliminary study, the best fit to perform image enhancement were the *Anisotropic Diffusion filter* ($\overline{X}_{\text{hydrate}}$ = 16.79 %) and the *Non-local means filter* ($\overline{X}_{\text{hydrate}}$ = 17.11 %), since both show good agreement in consistency and subsequent numerical estimation of the hydrate bulk. The final decision was to use both filters in a combined manner on the datasets in 3D mode which arose because neither the *AD* filter nor the *NLM* filter worked on all phases properly. The *AD* filter caused problems when targeting the quartz-gas phase boundary and the *NLM* filter hampered the segmentation of the quartz-

hydrate phase boundary. Both edge detection filters, *Sobel* (*EDS*) and *Moments* (*EDM*), were not suitable for the data since the subsequent segmentation could not be carried out.

## 2.4 Segmentation & Volume rendering

In this section, we describe the necessary steps of segmentation for this study. Those steps are crucial if the obtained 3D data will be used further as a numerical model input. This process results in the transformation of voxels of certain grey scale range.

Given that a histogram-based segmentation was impossible to conduct, we combined the watershed algorithm and a region growing technique to classify features like phase distribution (Sell et al., 2015) in the filtered sub-volumes of every full dataset (each 28 GB). Watershed algorithms are used to semi-automatically segment tomographic data, treating gray values of the



gradient image as a topographic map where each gray value stands for a specific altitude. After the placement of seeds or marker regions in catchment basins representing minima which are basically homogeneous areas with a low gradient, the image is virtually immersed. As soon as "water" from different basins meet a so-called watershed boundary is placed in-between (Vincent and Soille, 1991;Wang, 1997). The algorithm performs well placing watersheds on sharp intensity edges

showing steep gradients and producing maxima in the gradient image as discussed in detail in previous literature (Iassonov et al., 2009;Bleau and Leon, 2000). With this approach it was possible to extract the hydrate phase and the gas/water phase. The quartz phase caused major problems because at the rims of the grain appeared strong irregular edge enhancement effects that could not be removed by filtering (Figure 7).

As the enhanced edges were of almost the same grey value as the hydrate phase, masking was applied. First, the void and the hydrate were segmented using the watershed algorithm than the resulting binarized volume of two phases was used to mask the original data in order to extract the quartz phase which worked pretty well. Several arithmetic corrections including numerical reconstruction steps and morphological dilatation and erosion were performed to split phases before resulting in a satisfying binarized dataset. Initially every filtered dataset was segmented into three classes: water of gas-filled pores, quartz

and hydrates. A fully binarized dataset along the *AD-NLM* filtered data is shown in Figure 8. For the purpose of 3D visualization, we recommend to create a triangulated surface of the binary data (Figure 9) followed by the application of volume rendering (Figure 10). For this we used the *unconstrained smoothing* algorithm and the *compactify* function available in *Avizo*.

Due to the earlier mentioned low-density contrast issue between the gaseous and aqueous phase the segmentation of the residual water film was accomplished slice-by-slice for smaller ROIs using a *Region growing* function (Figure 11). For visualization in 3D a surface creation tool is applied prior the actual rendering.

## 2.5 Preliminary Study on the effect of image enhancement on numerical simulations

Note that the inability to fully characterize the microstructural details of the material under examination can lead to

disagreements between numerical estimates of mechanical properties based on SRXCT images and experimentally derived results (Sell et al., 2013b). Such microstructures are thought to significantly influence the effective elastic properties of a rock sample. Performing such study directly on the hydrate bearing sediments would have taken too many side effects into account as described earlier (including strong X-ray attenuation, liquid and hydrate phases). Consequently, we have decided to perform a preliminary study to predict the effect of image enhancement on subsequent modelling based on synchrotron data of a nearly

one-phase material. For details on this pre-study please refer to Saenger & Sell, 2013. The pre-study was done on a synchrotron scanned Berea sandstone data set, a well-known "standard" in CT studies and lab investigations (Andrä et al., 2013b, a). We



chose a cylindrical specimen of 3 mm diameter which was also scanned at the TOMCAT beamline of the Swiss Light Source at the Paul Scherer Institute in Villigen, Switzerland. The reconstruction process yielded in an image matrix of 1024 x 1024 x 1024 voxels, with a voxel size of 0.74 µm resulting from the selected tenfold magnification and a field of view of 1.5 x 1.5 mm. Also in this case artifacts from the scanning and reconstruction process have been found (e.g. noise, streaks, non-uniform

brightness and edge enhancement). These have been treated as discussed in section *Image Enhancement.* For further investigations a region of interest called *Ber1* cropped to 400 x 400 x 400 voxels was chosen from the open-source dataset *Case 3* which can be downloaded from SI of Madonna et al. (Madonna et al., 2013). For details on the sample *Case 3* please refer to the mentioned publication. Three of the earlier discussed image filters were applied to the original dataset Ber1. The *3D median filter* has been applied to Ber1a, the *Non-local means filter* to Ber1b and the *Anisotropic Diffusion filter* to Ber1c.

All datasets including the original unfiltered have been segmented using the semi-automatic watershed algorithm. Further numerical investigations were conducted including the determination of the total porosity *Phi*, the permeability *K* and the effective *P*-wave velocities. Undoubtedly the variation in image enhancement and subsequent determination of grain contact has a strong effect on the obtained rock properties.

The results showed that the effect of image enhancement leads to significant variation regarding the assigned grain contacts (Figure 12). The total porosities vary from 12 % for the unfiltered dataset Ber1 to 18 % for the dataset Ber1b where the image noise was reduced by applying the *NLM* filter. Hence the majority of the pores (>98%) was found to be connected on the basis of the open-closed porosity approach of the PoroDict module of GeoDict. The watershed algorithm has been applied on the samples (Ber1a-c) did not have any influence on the porosity nor the permeability tensor. Instead the number and trend of the

modeled grain boundaries by using the *binseparate* function revealed a strong impact on the effective *P*-wave velocities that varied from 3706 to 5043 m/s (Sell et al., 2013b) . All results are depicted in Figure 10, where a representative slice (no. 111) of each case is shown, revealing the effect of filtering and assignment of grain boundaries, as well as the rock characteristics. Comparing the *P*-wave velocity, porosity and permeability sustained by digital rock analysis with results from lab investigations performed on this Berea sample (Madonna et al., 2012) we found the best matching dataset to be Ber1b filtered

with *NLM*. Digital rock analysis indicated a total porosity of 18% and a permeability of 150 mD for Ber1b. The worst matching dataset during the determination of effective physical properties were the unfiltered and the median filtered one with a total porosity of 12% and a permeability of 80 mD (Sell et al., 2013b) .

## 3. Wave propagation modelling in hydrate-bearing sedimentary matrices

In this section, we present a first application of our derived post-processing data to wave propagation modelling. A series of

numerical experiments based on segmented 3D images of the structure has been conducted to obtain the p-wave velocity trend with in- or decrease of the hydrate bulk. This has been done to validate our model approach to aim for further wave propagation modelling including s-wave velocity and attenuation in future studies. As mentioned earlier, methane hydrate has been



substituted by Xe-hydrate that exhibits far reaching similarities in crystal structure and molecular size of guests (Table 1). Yet, in terms of the compressibility these clathrates are noticeably different; structures hosting Xe gas are about 10% stiffer than in case of $CH_4$ (Hansen et al., 2016). The difference originates in guest-host coupling which is strong for tightly fitting Xe and weaker for somewhat smaller methane that freely rattles in cages (Chazallon et al., 2002;Schober et al., 2003). Another

potential difference might be found in the crystal size for both clathrates due to somewhat different density of nucleation sites. Yet, without a definite proof we must assume for the moment that the hydrate nucleation and crystallization processes of both clathrates in the sedimentary matrix are comparable. We also take digitized Xe-hydrate distribution and microstructure obtained from SRXCT and assign to it elastic moduli of $CH_4$ hydrate: bulk modulus ($K$), shear modulus ($G$), P-wave modulus ($M$) and the density ($Rho$).

We applied the rotated staggered finite-difference method appropriate for dynamic measurements. The fundamental idea of the model is discussed in detail by (Saenger et al., 2004). This approach studies the wave propagation in heterogeneous materials within the long wavelength limit. Using this method it is possible to predict precisely effective elastic properties of various pore geometries in the relatively wide range between the upper and lower Hashin-Shtrikman bound. Complex

structures, as observed in our samples, trigger strong scattering which can be treated only by numerical techniques since an analytical solution of the wave equation is not available. *Finite difference* (*FD*) methods discretize the wave equation on a grid and replace spatial derivatives by *FD* operators using neighboring points. Instability problems on a staggered grid may be caused by discretization when the medium contains high contrast discontinuities (e.g., pores or fractures). These difficulties can be avoided by using the rotated staggered grid technique proposed by Saenger and Bohlen, 2004. Since the *Finite*

*differences* approach is based on the wave equation without physical approximations, the method accounts not only for direct waves, primary and multiply reflected waves, but also for surface waves, head waves, converted reflected waves, and diffracted waves (Saenger et al. 2004).

Furthermore, the application of this method has already been benchmarked (Andrä et al., 2013a). On top of the 3D rock model

a body force plane source is applied using a homogeneous buffer zone of assigned vacuum (Sell et al. 2015). The plane wave travels through the embedded digitized 3D sample volume (Figure 13.1). Two plane receivers on the top and the bottom of the model measure the time-delay of the plane wave's peak amplitude caused by the inhomogeneity of the rock (Madonna et al. 2013). As the model input is chosen to a volume of 400 x 400 x 400 pixels due to computational limits, the segmented data sets needed to be reduced in size. This was accomplished by using the *Resample* tool based on the *Lanczos* filter implented in

*Avizo*. Usually the default kernel *Lanczos* yields sufficiently good results for both minifications and magnifications. In the present case a cropped dataset of 1800 x 1800 x 1800 voxels with a voxel size of 14.06 was reduced to a cube of 400 x 400 x 400 with a voxel size of 4.5.



Effective velocities of the hydrate bearing sedimentary matrices are determined by comparing the simulated results with those of a reference model. Note that the quartz grains have not been compacted during sample preparation wherein the hydrate is located in the pore space which affects the structure to be stiffer with an increase in hydrate saturation. (Figure 13.2 *and* 13.3).

As the stop-and-go scanning procedure (Chaouachi et al., 2015;Falenty et al., 2015) permits a time-resolved observation of hydrate formation, the numerically obtained *p*-wave from the scans confirm the expected trend of an increase of p-wave velocities with higher hydrate bulk in the sediment (Priest et al., 2009). For each post-processed scan the *p*-wave velocity has been determined following the workflow described in this paper. In Figure 13.2 the p-wave velocity simulated at the first time step is visualised for a sample with 7% of hydrate. The simulated *p*-wave velocity for a sample containing 17% of hydrate is visualised in Figure 13.3. The comparison of both figures shows clearly how the *p*-wave velocity increases due to a higher hydrate bulk as the GH holds the quartz in place and stiffens the structure.

In Figure 14, the results of the simulated $V_p$ for series of 8 scans is depicted. In absence of hydrate (*scan 5*) the lowest p-wave velocity of 1462 ms$^{-1}$ is given for the model series. This p-wave value is typical for partially water-saturated sands which is also reasonable. The highest p-wave velocities have been obtained for the full-transformed states as expected (*scans 1 & 8*). Scan 1 was taken when hydrate was formed from the free-gas system starting at the gas-water interface and scan 8 when hydrate was formed from a system where liquid water is metastable enriched with a gas (gas-enriched system). For the first case the obtained p-wave velocity is 2368 ms$^{-1}$ and for the second case it is 2408 ms$^{-1}$. Therefore, at the full-transformed state of hydrate formed in a free-gas system (*scan 1*) the p-wave velocity is slightly lower than at the full-transformed state of hydrate formed in a gas-enriched system (*scan 8*) which has been discussed earlier in literature (Priest et al., 2005;Konno et al., 2015;Waite et al., 2009). With the presented results of the modeled p-wave velocities the model approach is in a realistic range when compared with field (Carcione and Gei, 2004;Riedel et al., 2002;Yuan et al., 1996) and laboratory data (Zhang et al., 2011;Priest et al., 2009;Priest et al., 2005);  it is noteworthy, that the modelled results give slightly lower values than the experimental results, yet this may well be explained by the significantly lower hydrate saturation (<20%) compared to results from laboratory work. Furthermore, one should take into account that investigations on seismic responses conducted in laboratory often use pre-compacted samples (Matsushima et al., 2015) which we, in comparison, did not but unconsolidated samples. Moreover, there are uncertainties with the elastic response of Xe-hydrate, likely to be different from the one of CH$_4$-hydrate, which was taken for our analysis.

**Conclusions**

A novel processing approach, based on tomographic data with sub-µm scale resolution, is presented which allows to obtain the effective elastic properties of hydrate bearing sediments. The data processing is requiring preliminary studies to aim for best possible image quality enhancement. Applying the proposed method it is possible to approximate the seismic velocities of sediments containing less than 20% of hydrate within a realistic range at the sub-µm scale. As the time-resolved images



produced a time series of hydrate formed from different scenarios (free-gas and gas-enriched) we are also able to distinguish between those cases in terms of seismic response. We recommend conducting preliminary studies prior to the processing of 3D tomography data to evaluate the image enhancement workflow. Future outlook is to focus on the *S*-wave velocities and attenuation since the presence of a thin fluid layer observed in the structures could explain the anomalous seismic responses

5   of hydrate bearing-sediments observed in field studies. For this, a satisfying solution to segment the thin fluid layer is needed.





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

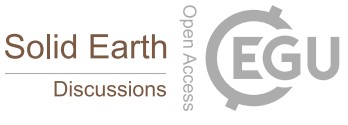

14 Figures

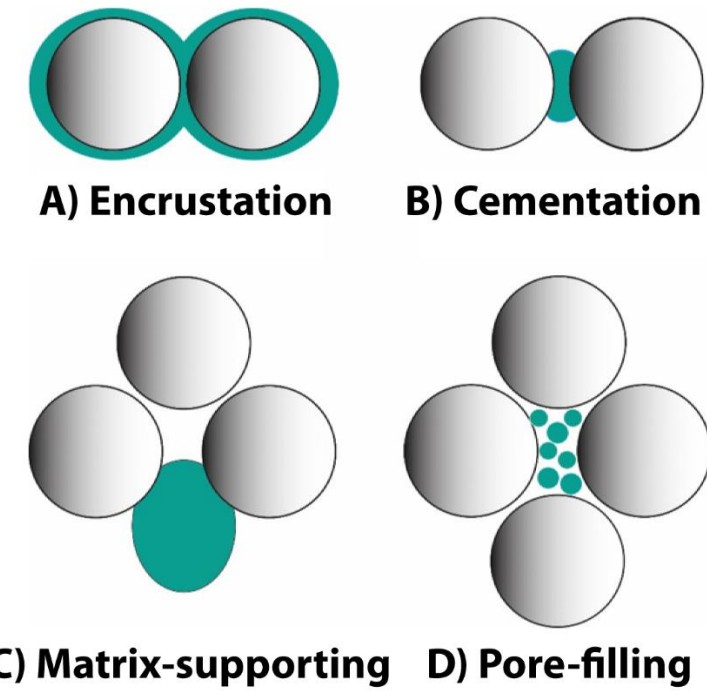

**Figure 1:** Various habits of how hydrate might be located in the sedimentary matrix as discussed in previous studies: Encrustation, Cementation, Matrix supporting and Pore-filling; modified after(Dai et al., 2004)





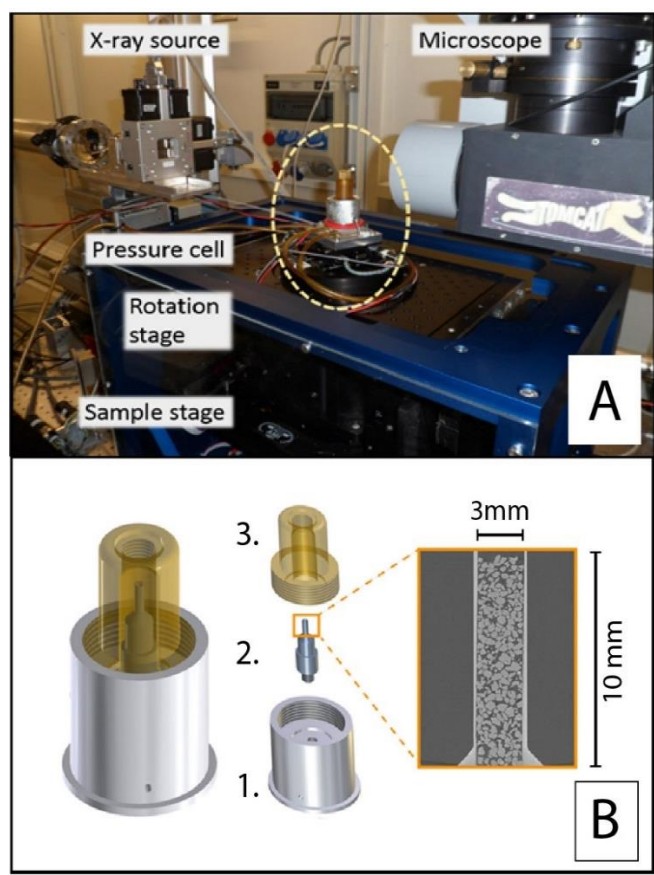

**Figure 2: A) Experimental setup at the TOMCAT beamline. B) The pressure cell consisting of the base [1], the sample holder [2] and the dome [3].**




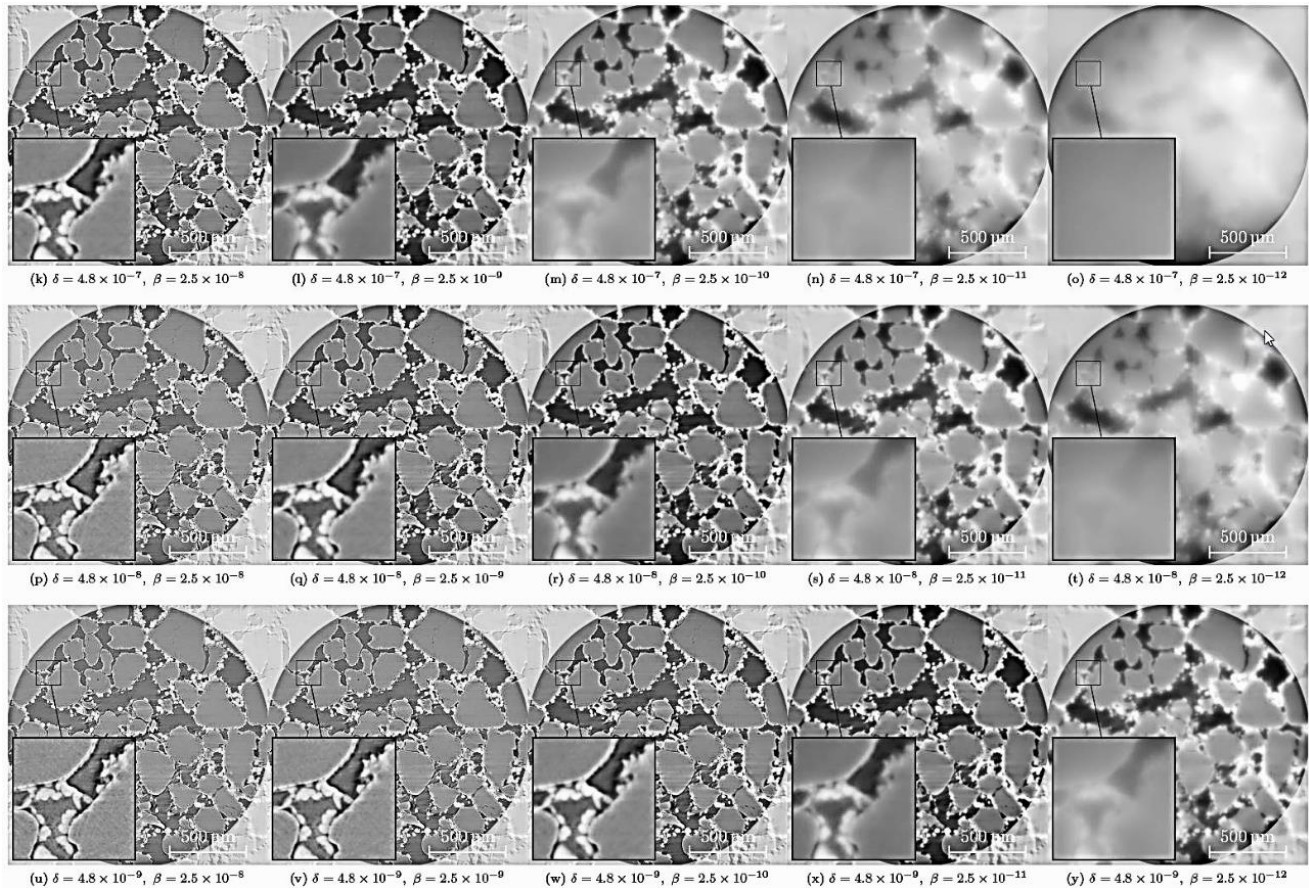

**Figure 3: Selected results of the approach to enhance the contrast between the gaseous and aqueous phase by applying the reconstruction algorithm of Paganin on a 2560 x 2560 voxel 2D slice. The insets show a 75 x 75 pixel sized Region of Interest, magnified 5x.**





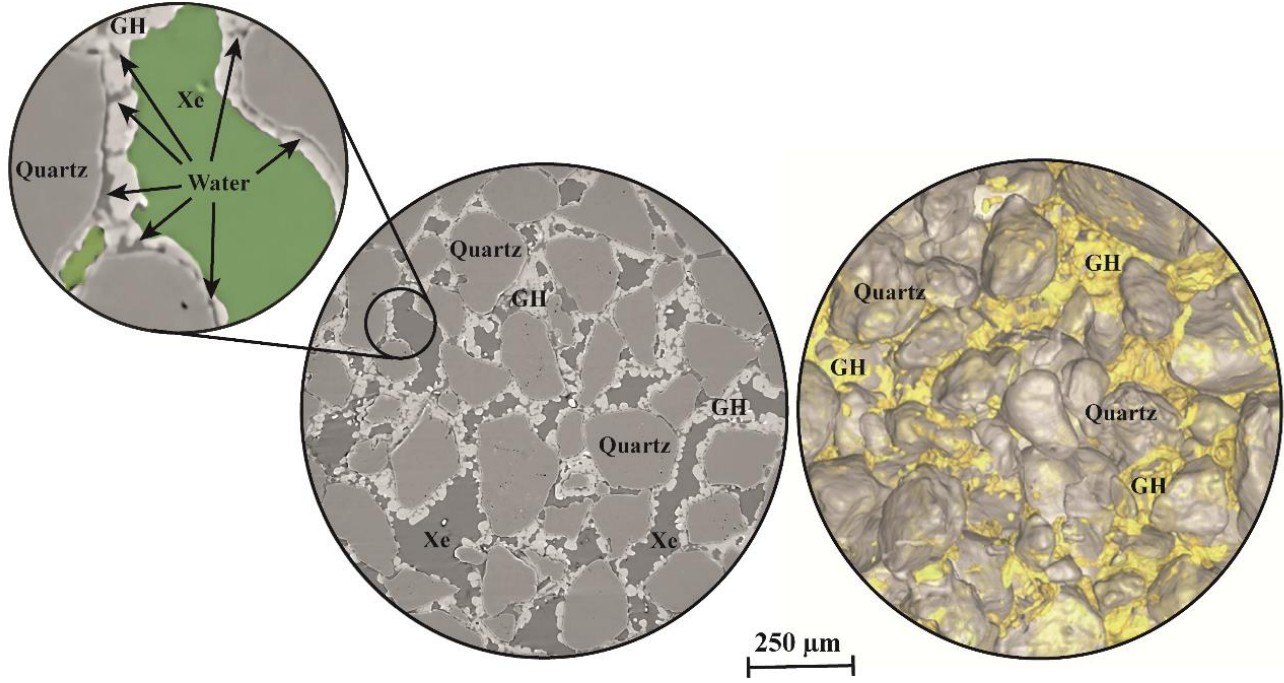

**Figure 4: Overview of a 2D slice in XY-direction and the equivalent 3D rendered volume. The presented sample contains 17 Vol% hydrate. The 2D slice has a dimension of 2560 x 2560 pixels with a pixel size of 0.76 µm². The volume rendered image depicts the quartz grains in grey and the hydrate in yellow – this dataset is suitable for further model investigations. The zoom-in depicts all the**
5  **phases present in the samples: Xenon - Xe (green), Quartz grains, Gas hydrate - GH and water.**




**Figure 5: Selected filters applied to the original image to enhance the quality for further segmentation steps. Also, the results of the numerical estimation of hydrate content with the PoroDict module of GeoDict after arithmetic correction are depicted in the upper right corner. Blank circles indicate poor image quality where segmentation was not accomplishable.**




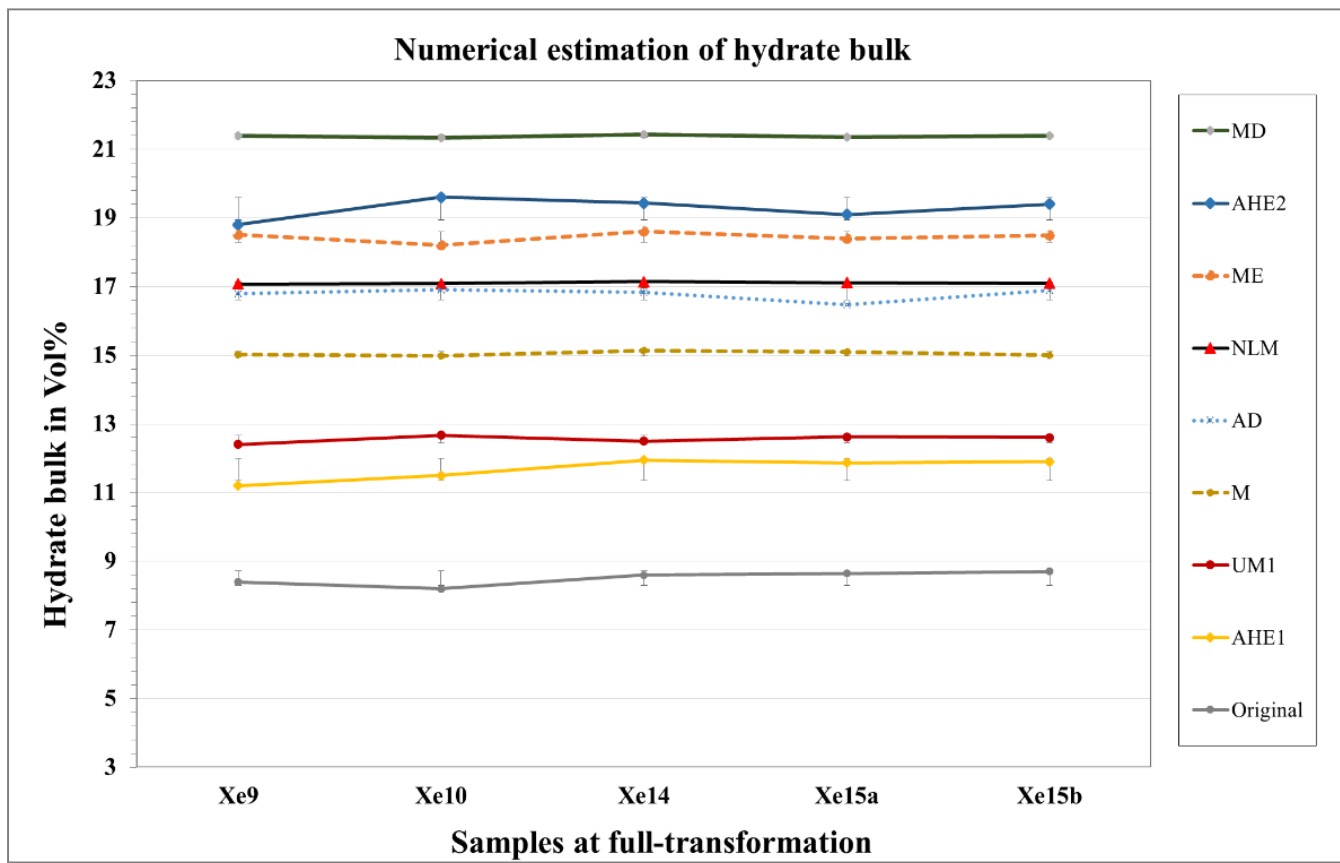

**Figure 6: Diagram of the results obtained from numerical estimation of the hydrate content in 5 different samples containing approx. 17% controlled by the water available at the initial formation stage.**




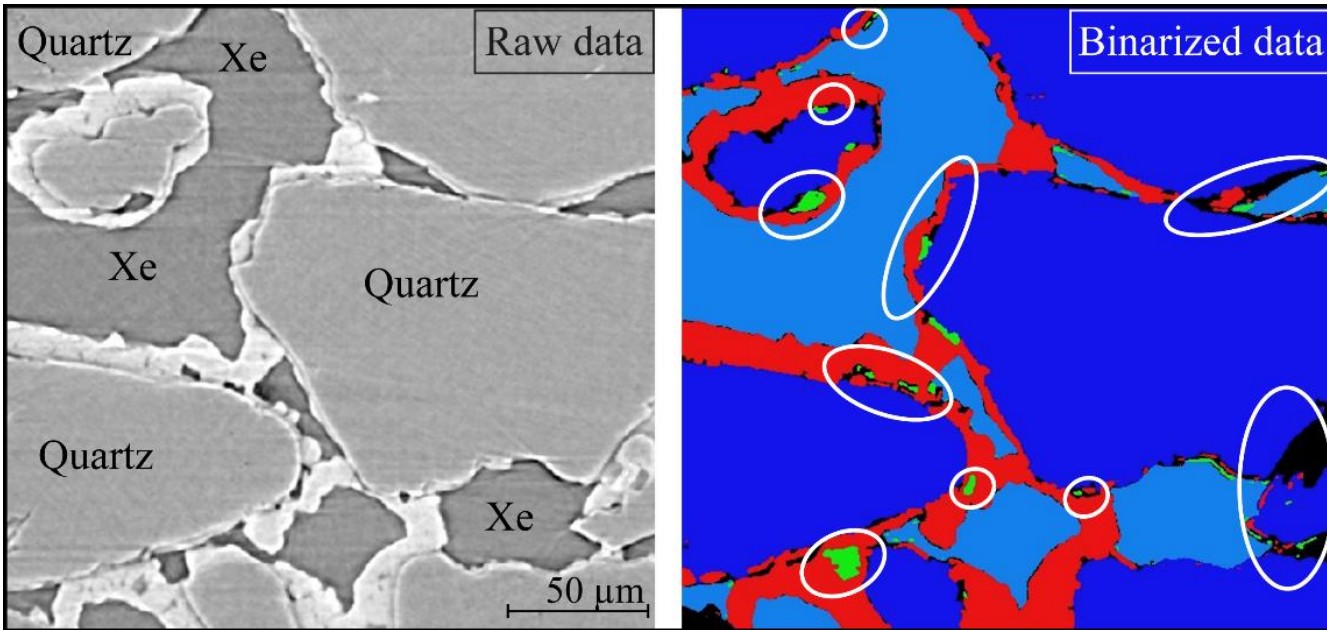

**Figure 7: Segmentation problems occurring in samples of low-grade image enhancement quality.**





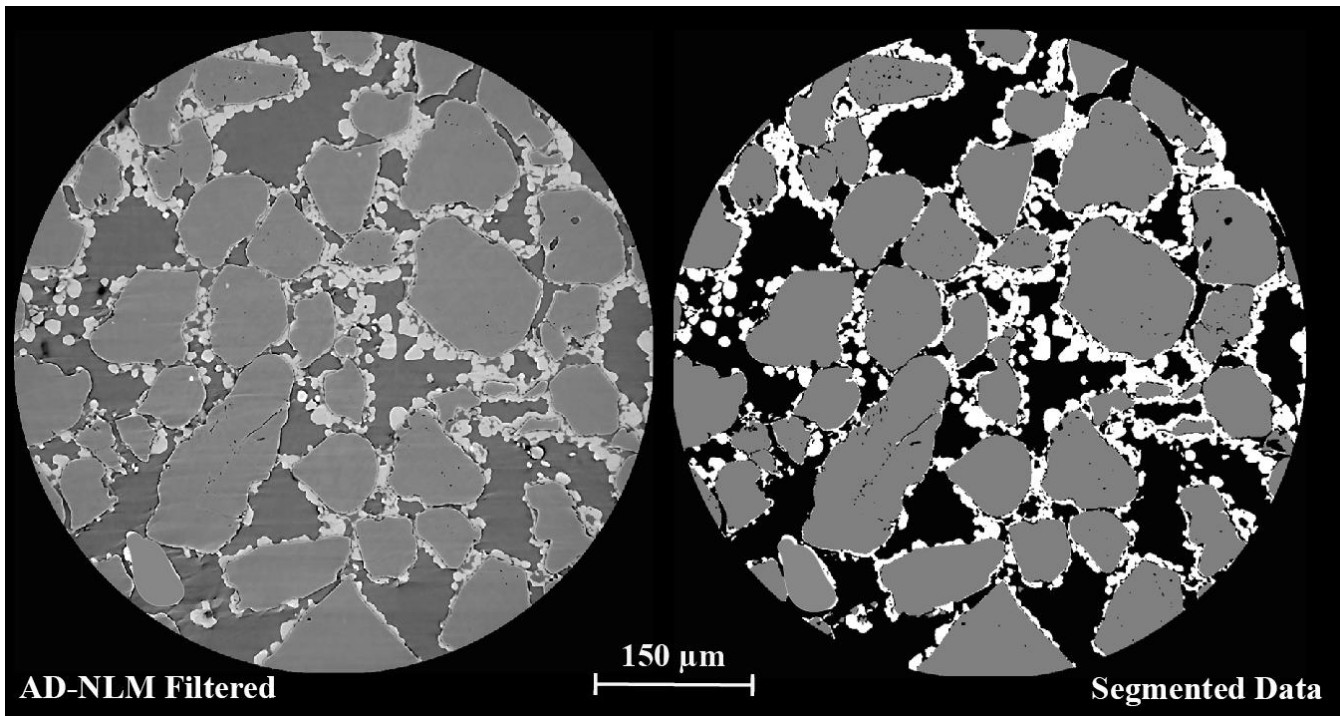

**Figure 8: Example of a successfully segmented dataset. On the left a representative 2D slice in XY direction and the resulting binarized image. Size of the 2D slice is 2560 x 2560. Black = pore space, gray = quartz and white = hydrate. This segmented data is ready to be utilized as a direct model input.**




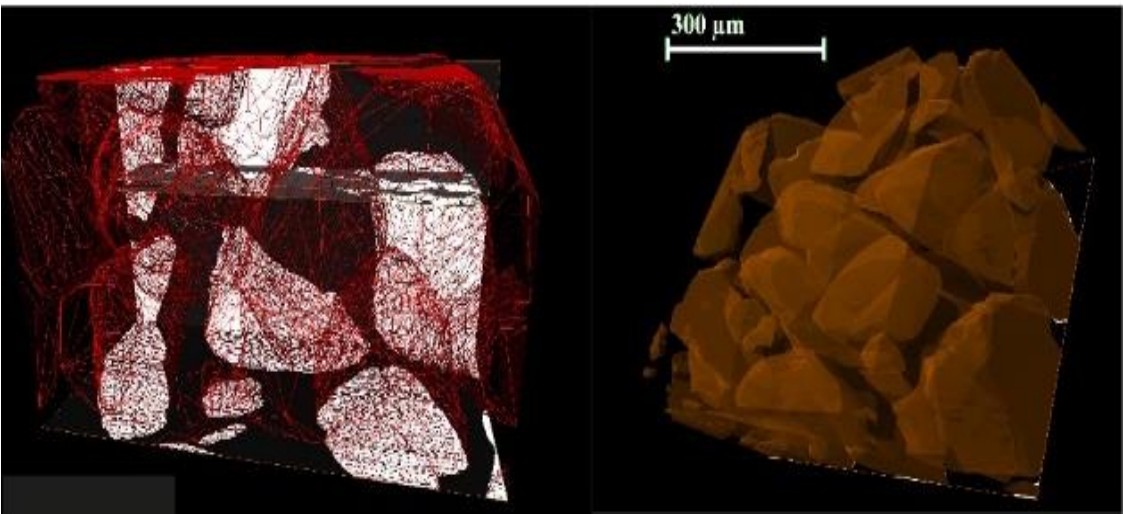

**Figure 9: Surface triangulation of quartz grains and the surface view on α-scale transparency.**



**Figure 10: Series of hydrate formation visualized in 3D. For this a ROI of 400 x 400 x 400 was picked. 10.1 Initial state where no hydrate is present; 10.2 & 10.3 Hydrate formation starts and proceeds from a free-gas environment; 10.4 Full-transformation state; 10.5 Full-transformation state embedded in the sedimentary matrix. Centered figure shows all hydrate formation steps within one image where some steps are set on α-scale transparency to gain information on hydrate structures beneath.**





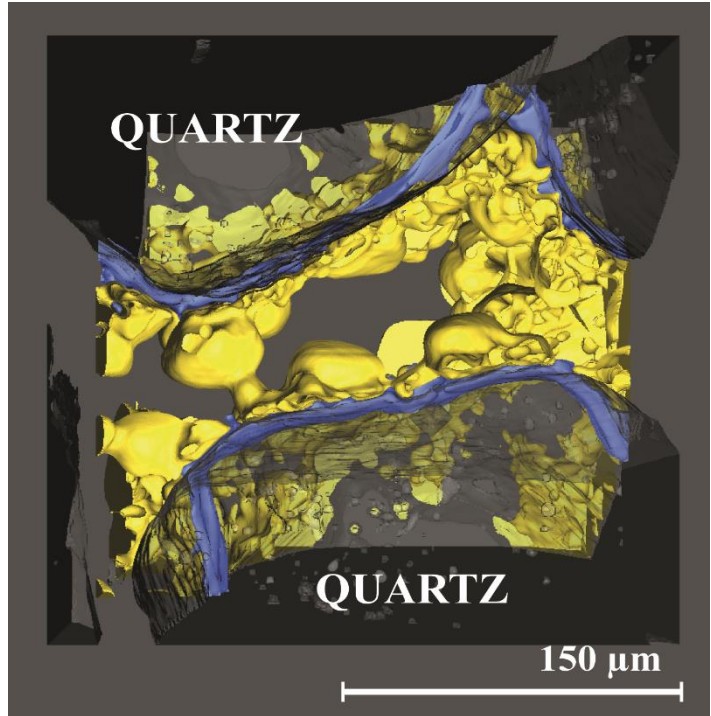

**Figure 11: Volume rendered image of the water film (blue) in-between hydrate (yellow) and the quartz grains - for better understanding the outer surface of the grains are depicted at α-scale transparency. Voxel resolution for this scan was 0.38µm.**



**Figure 12:** This figure comprises the results of the preliminary study on the effect of filtering on the modelling of effective elastic properties. Representative 2D slices (#111 of the XY plane) are depicted



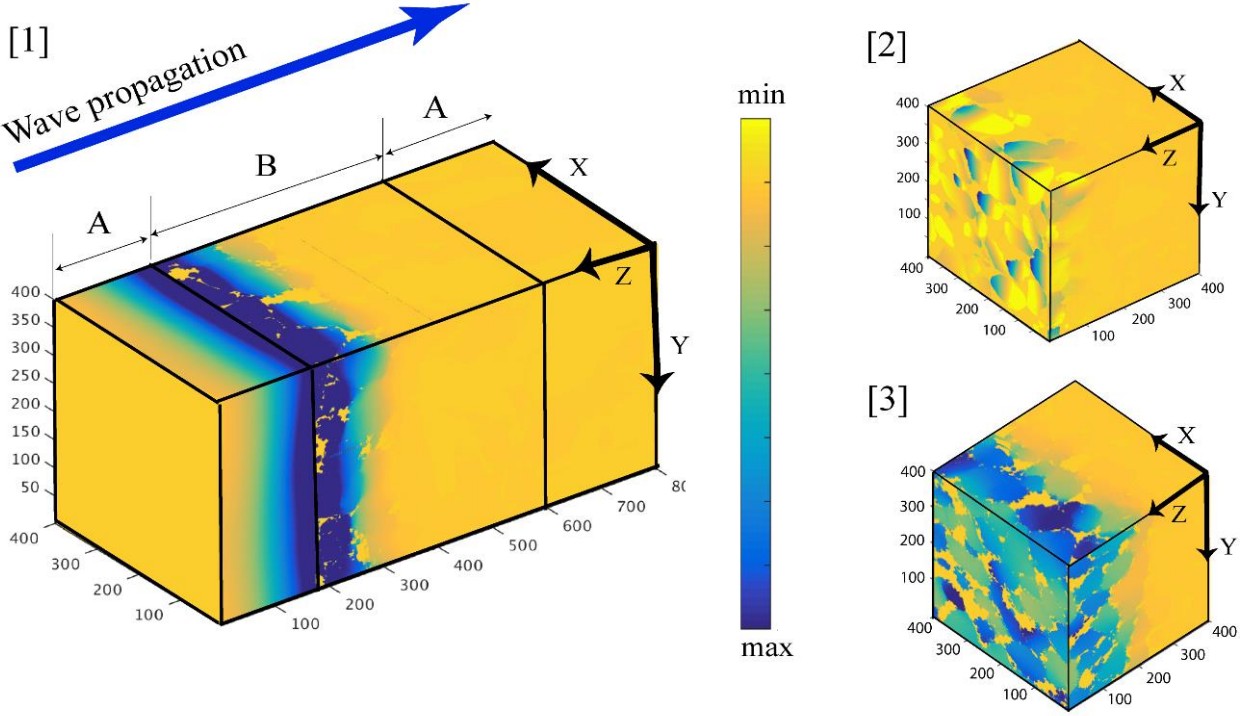

**Figure 13: [1.] Visualization of the RSG model where A is a predefined homogenous solid in which the 400 x 400 x 400 sized volume is embedded. The plane wave propagates in direction of the arrow. [2.] Visualization of the Vp in a sample containing roughly 7% of hydrate. [3.] Visualization of the Vp in a sample bearing 17% of hydrate where the Vp is significantly higher due to the increased hydrate bulk.**




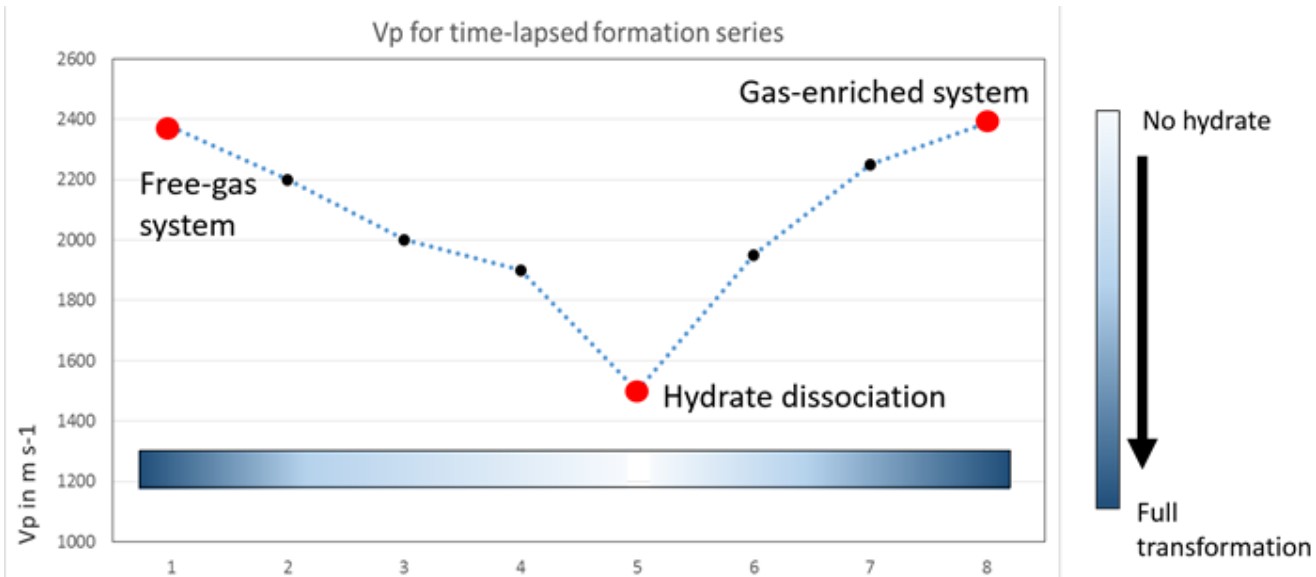

**Figure 14: Results of the p-wave modelling of a Series of scans starting at the initial state forming hydrate from a free gas system up to the full-transformed state, followed by a controlled destabilization of the system to maintain hydrate dissociation from which again the hydrate formation was triggered up to a full-transformed state in a gas-enriched were conducted.**





5    **Table 1: Comparison of crystallographic properties of xenon- and methane hydrate (Hansen et al., 2016). Second part of the table depicts the elastic moduli latter assigned for modelling the effective elastic properties. Values of M (P-wave modulus), K (Bulk modulus) and G (Shear modulus) after (Helgerud et al., 2003).**

|  | Xe-hydrate | CH4-hydrate | Quartz |
|---|---|---|---|
| Hydrate structure | sI | sI |  |
| Cage diameter | 5.09 - 5.86 | 5.1 - 5.85 |  |
| Lattice constant at 276K [Å] | 11.988 | 11.991 |  |
| Molecular size | 4.58 | 4.36 |  |
| M [GPa] | Substitution of elastic modulis and density for modeling | 13.57 | 96.98 |
| K [GPa] |  | 8.76 | 37.8 |
| G[GPa] |  | 3.57 | 44.3 |
| Rho [kg m-3] |  | 982 | 2648 |

