# Peer review of "On the path to the digital rock physics of gas hydrate bearing sediments – processing of in-situ synchrotron-tomography data"

_Solid Earth, 2016_

## Referee Comment (RC1) · Anonymous Referee #1 · 13 Apr 2016

This study claims a post-processing procedure for the synchrotron tomographic data, involving the density contrast enhancement, as well as 2D and 3D phase rendering and segmentation; of particular interest is their application in the physical parameters simulation with the help of the derived results. Various filters for the image enhancement are illustrated and compared, which depicts a rather beautiful picture of the differences among them, and thus would help a lot for the readers to figure out a proper one. This will contribute considerably to the following studies. Volume rendering is always an ambiguous issue in 3D image processing, particularly in studies on gas hydrate where low density contrast phases such as water and hydrates are frequently present. The method introduced here by a combination of watershed algorithm and a region growing

technique appears practical, yet still some problems remain, like the effect of signal-noise ratio on the accuracy of the region growing method. The simulation of physical properties through the digital rock data is a quite interesting topic, which could on one hand predict what is difficult to measure, and also explain the overall property from a much more microscopic angle of view on the other hand. The velocity of P-wave propagation in sediments is shown and further calculation on conductivity (thermal, hydraulic, electrical) or mechanical strength could likely be expected as well. Overall, this study is worth a positive comment, from which I believe the readership will benefit a lot.

---

## Short Comment (SC1) · 12 May 2016

The authors describe pore scale imaging of gas hydrates in sediments and the processing of the obtained data and their use for numerical modelling. Especially the detailed explanation of image processing to yield a 3D segmentation of the sample components is of great value for future imaging studies of rock samples with and without gas hydrates. Further, it is a great addition to other publications of this dataset which focus more on the experimental setup (Chaouachi et al., 2015). Some questions remain about the numerical modeling section: Your images indicate that the hydrate does not cement the grains however your modelled P-wave velocities increase a lot for low hydrate saturations indicating a significant stiffening of the sediment which –

according to effective medium models – would only be achieved by cementing the grain contacts. Comparing your numerical modeling results to effective medium models (Dvorkin, Helgerud, Ecker) and laboratory data (by Priest, Kneafsey, Waite, etc.) might be a good benchmark for your numerical modelling results. This section could benefit from a more thorough discussion about factors causing differences between modelled and laboratory / field data. The authors conclude that this study enables to distinguish gas hydrate from a gas enriched system and gas hydrate from a free gas system based on their seismic response. However, the modelled velocities for both differ by just 40 m/s. That would actually indicate that both formation mechanisms lead to similar elastic properties. . Observed differences from the pore-scale imaging of these two gas hydrate types are not reported in the study. The study is certainly worth publishing but could be improved by relating modelling results to published velocity data.

---

## Short Comment (SC2) · 16 May 2016

Dear Anonymous Reviewer #1,

the authors of the method article "On the path to the digital rock physics of gas hydrate bearing sediments – processing of in-situ synchrotron-tomography data" were pleased to receive the positive comment and feedback regarding the submitted discussion paper from you.

Thank you for taking the time to read and comment on our article.

Best,

Kathleen Sell (corresponding author) and colleagues

---

## Short Comment (SC3) · 21 May 2016

Line 20: "relies" instead of "relays"

Line 22: "has been interpreted before" instead of "has been earlier interpreted"

Line 27: I'm not quite sure what you mean by "habits" – it also occurs in the figure caption of Figure 1. Can you maybe rephrase it? E.g. hydrate distribution

Line 4: "systems" instead of system

Line 12: "these" instead of "this" and "met" instead of "meet"

Line 14: I'm not sure what you mean by "refractory" here (that may be my lacking knowledge of the terminology though)

Line 24: instead of "followed" you could use "monitored" or "observed"

You're mentioning the density differences between water and methane hydrate vs. Xenon hydrate. I think it would be helpful to include a table with densities for your sample components (grains, Xe hydrate, Xe gas, water)

In the first paragraph it sounds like you're saying that Xenon gas and water could not be distinguished (and it sounds like that again on Page 8, line 14) yet in the second paragraph (line 11) you say you were able to distinguish between gas and water. Can you clarify this?

Line 11: "commonly occurred" instead of "occurred most"

Line 26: "prior to" instead of "prior"

Line 14: "number of iterations" instead of "number of iteration"

Line 3: "meets" instead of "meet" and a comma comma after "meets"

Line 11: "then" instead of "than"

Line 14: "water or gas-filled" instead of "water of gas-filled"

Line 2: just "yielded" instead of "yielded in"

Line 23 "obtained" instead of "sustained"

Paragraph 2: I think it would be helpful if you mentioned the laboratory derived values by for porosity and permeability by Madonna et al. to compare to your numerically derived values

Page 10:

Line 21: "multiple" instead of "multiply"

Line 24: Can you elaborate a bit more of what you mean by "benchmarked" here? Did they compare modelled and laboratory measured data?

Line 31/32: voxel sizes are missing units

Page 11/12

Note: most of the things I mentioned in my short comment are referring to these pages. What makes me wonder most is that your increase in velocities is really high for a hydrate saturation of 17% (for both tested formation mechanisms). So would you conclude that the hydrate significantly stiffens the sediment even though the hydrate does not appear to actually touch the grains? You could add some more detail here to your discussion. I think this is actually really interesting! Your images show that the hydrate doesn't follow any of the end-member models discussed by effective medium theory (pore filling, contact cementing etc.). So maybe one of your conclusions could also be that we need better physical models than the effective medium ones.

It would also be interesting to discuss whether your images showed any differences in hydrate distribution for the two formation mechanisms (from free gas + water and from gas enriched water). The literature usually assumes the first one forms cementing hydrate while the latter forms pore filling hydrate. It seems like you didn't observe this difference – neither in your images nor your modelled velocities. That's an interesting observation and worth discussing!

Line 17 to 21: If I'm not mistaken the conclusion from the studies your mentioning (Waite, Priest etc.) is usually that the velocity is higher for hydrate formed from free gas and water than for hydrate formed from gas-enriched water (especially at low saturations, like the ones you used for your experiments). Your model indicates something different. You could add some phrases to hypothesize why your model results differ from lab data.

---

## Referee Comment (RC2) · M. Schindler (Referee) · 26 May 2016

The authors describe pore scale imaging of gas hydrates in sediments and the processing of the obtained data and their use for numerical modelling. Especially the detailed explanation of image processing to yield a 3D segmentation of the sample components is of great value for future imaging studies of rock samples with and without gas hydrates. Further, it is a great addition to other publications of this dataset which focus more on the experimental setup (Chaouachi et al., 2015). Some questions remain about the numerical modeling section: Your images indicate that the hydrate does not cement the grains however your modelled P-wave velocities increase a lot for low hydrate saturations indicating a significant stiffening of the sediment which –

according to effective medium models – would only be achieved by cementing the grain contacts. Comparing your numerical modeling results to effective medium models (Dvorkin, Helgerud, Ecker) and laboratory data (by Priest, Kneafsey, Waite, etc.) might be a good benchmark for your numerical modelling results. This section could benefit from a more thorough discussion about factors causing differences between modelled and laboratory / field data. The authors conclude that this study enables to distinguish gas hydrate from a gas enriched system and gas hydrate from a free gas system based on their seismic response. However, the modelled velocities for both differ by just 40 m/s. That would actually indicate that both formation mechanisms lead to similar elastic properties. . Observed differences from the pore-scale imaging of these two gas hydrate types are not reported in the study. The study is certainly worth publishing but could be improved by relating modelling results to published velocity data.

Page 2 Line 20: "relies" instead of "relays" Line 22: "has been interpreted before" instead of "has been earlier interpreted" Line 27: I'm not quite sure what you mean by "habits" – it also occurs in the figure caption of Figure 1. Can you maybe rephrase it? E.g. hydrate distribution Page 3 Line 4: "systems" instead of system Line 12: "these" instead of "this" and "met" instead of "meet" Line 14: I'm not sure what you mean by "refractory" here (that may be my lacking knowledge of the terminology though) Line 24: instead of "followed" you could use "monitored" or "observed" Page 4 You're mentioning the density differences between water and methane hydrate vs. Xenon hydrate. I think it would be helpful to include a table with densities for your sample components (grains, Xe hydrate, Xe gas, water) Page 5 In the first paragraph it sounds like you're saying that Xenon gas and water could not be distinguished (and it sounds like that again on Page 8, line 14) yet in the second paragraph (line 11) you say you were able to distinguish between gas and water. Can you clarify this? Line 11: "commonly occurred" instead of "occurred most" Line 26: "prior to" instead of "prior" Page 6 Line 14: "number of iterations" instead of "number of iteration" Page 8 Line 3: "meets" instead of "meet" and a comma comma after "meets" Line 11: "then" instead of "than" Line 14: "water

or gas-filled" instead of "water of gas-filled" Page 9 Line 2: just "yielded" instead of "yielded in" Line 23 "obtained" instead of "sustained" Paragraph 2: I think it would be helpful if you mentioned the laboratory derived values by for porosity and permeability by Madonna et al. to compare to your numerically derived values Page 10: Line 21: "multiple" instead of "multiply" Line 24: Can you elaborate a bit more of what you mean by "benchmarked" here? Did they compare modelled and laboratory measured data? Line 31/32: voxel sizes are missing units Page 11/12

Note: most of the things I mentioned in my short comment are referring to these pages. What makes me wonder most is that your increase in velocities is really high for a hydrate saturation of 17% (for both tested formation mechanisms). So would you conclude that the hydrate significantly stiffens the sediment even though the hydrate does not appear to actually touch the grains? You could add some more detail here to your discussion. I think this is actually really interesting! Your images show that the hydrate doesn't follow any of the end-member models discussed by effective medium theory (pore filling, contact cementing etc.). So maybe one of your conclusions could also be that we need better physical models than the effective medium ones. It would also be interesting to discuss whether your images showed any differences in hydrate distribution for the two formation mechanisms (from free gas + water and from gas enriched water). The literature usually assumes the first one forms cementing hydrate while the latter forms pore filling hydrate. It seems like you didn't observe this difference – neither in your images nor your modelled velocities. That's an interesting observation and worth discussing! Line 17 to 21: If I'm not mistaken the conclusion from the studies your mentioning (Waite, Priest etc.) is usually that the velocity is higher for hydrate formed from free gas and water than for hydrate formed from gas-enriched water (especially at low saturations, like the ones you used for your experiments). Your model indicates something different. You could add some phrases to hypothesize why your model results differ from lab data.

---

## Referee Comment (RC3) · Anonymous Referee #3 · 3 Jun 2016

This manuscript details the processing of x-ray scans of hydrate-bearing sands to derive high-resolution 3-D CT representations of the pore-scale geometry, which can then subsequently be used in numerical models to better characterize the geophysical properties of these materials. This manuscript has the potential to be of great value in terms of improving 3-D imaging of rock samples as well as improving our understanding of hydrate-soil interaction at the pore-scale. However, there are a number of clarifications that I feel need to be addressed to ensure the significance of this paper.

The main clarification I feel is required is related to Section 3. Further details are required to describe the underlying constitutive model that is considered in the numerical model. As noted by the other referee there are a number of rock physics models that

have been developed to describe the interaction of the hydrate and soil grains, which give rise to significant differences in wave velocity at the reasonably low hydrate saturations used in your modeling. The wave propagation would depend not only on the respective elastic moduli of the components in the system but how these components interact. Details of the modeling, as in size of model, mesh size, how the discrete nature of particles and hydrate are modeled is lacking. These issues need to be addressed for this aspect of the paper to be worthwhile.

In addition there appears to be some ambiguity to exactly what phases are present in the samples tested. Your terminology of free-gas system and gas-enriched system does not clearly define the percentage of phases. Were both systems 'excess gas'? or are you inferring the 'gas enriched' is 'excess water'? Given the low velocity contrast between the two methods are they forming hydrate at the same location (relative to water) and therefore the minor variations in velocity are related to just reforming characteristics.

A final ambiguity I feel that needs to be addressed is the link between the title of this manuscript and the overall thrust of the paper. The title suggest that the focus of the paper in on the processing of the synchrotron data, however the conclusions focus nearly entirely on the success of the numerical modeling in determining seismic response.

I have attached an annotated manuscript highlighting some suggested grammatical corrections and areas where the grammar should be improved/revised.

Please also note the supplement to this comment:
http://www.solid-earth-discuss.net/se-2016-54/se-2016-54-RC3-supplement.pdf

**Supplement:**

[revised manuscript text omitted]

---

## Author Comment (AC1) · 20 Jul 2016

**On the path to the digital rock physics of gas hydrate bearing sediments –
Processing of in-situ synchrotron-tomography data**

Kathleen Sell[1], Erik H. Saenger[2], Andrzej Falenty[3], Marwen Chaouachi[3], David Haberthür[4], Frieder Enzmann[1], Werner F. Kuhs[2] & Michael Kersten[1]

[1] Institute of Geosciences, Johannes Gutenberg-University, Mainz, Germany
[2] International Geothermal Centre & Ruhr University, Bochum, Germany
[3] GZG Crystallography, Georg-August-University, Göttingen, Germany
[4] Now: University of Bern, Switzerland (former: SLS, Paul-Scherrer Institute, Villigen, Switzerland)

*Correspondence to:* Kathleen Sell (sell@uni-mainz.de)

*Please note: All our responses to remarks of Reviewers are given in italics.*

**Dear Reviewers,**

We appreciate the time, interest and effort you invested to evaluate our manuscript In the following, we respond to your questions, comments and concerns in order of appearance, to improve our manuscript based on your valued input.

Side note: Corrections of grammar and spelling will be additionally elaborated by the *Solid Earth Publication Board* prior publication.

Kind Regards

Kathleen Sell & Co-authors
* * *
**Anonymous Referee #1**

This study claims a post-processing procedure for the synchrotron tomographic data, involving the density contrast enhancement, as well as 2D and 3D phase rendering and segmentation; of particular interest is their application in the physical parameters simulation with the help of the derived results. Various filters for the image enhancement are illustrated and compared, which depicts a rather beautiful picture of the differences among them, and thus would help a lot for the readers to figure out a proper one. This will contribute considerably to the following studies. Volume rendering is always an ambiguous issue in 3D image processing, particularly in studies on gas hydrate where low density contrast phases such as water and hydrates are frequently present.

The method introduced here by a combination of watershed algorithm and a region growing technique appears practical, yet still some problems remain, like the effect of signal-noise ratio on the accuracy of the region growing method.

*Authors: The combined manner is the following: the watershed algorithm was used to segment the phases: hydrate, water/gas and the region growing technique to segment the grains. Due to this there are no signal-noise ratio effects expected.*

The simulation of physical properties through the digital rock data is a quite interesting topic, which could on one hand predict what is difficult to measure, and also explain the overall property from a much more microscopic angle of view on the other hand. The velocity of P-wave propagation in sediments is shown and further calculation on conductivity (thermal, hydraulic, electrical) or mechanical strength could likely be expected as well. Overall, this study is worth a positive comment, from which I believe the readership will benefit a lot.
* * *
**Referee #2: Mandy Schindler**

The authors describe pore scale imaging of gas hydrates in sediments and the processing of the obtained data and their use for numerical modelling. Especially the detailed explanation of image processing to yield a 3D segmentation of the sample components is of great value for future imaging studies of rock samples with and without gas hydrates. Further, it is a great addition to other publications of this dataset which focus more on the experimental setup (Chaouachi et al., 2015). Some questions remain about the numerical modeling section:

Your images indicate that the hydrate does not cement the grains however your modelled P-wave velocities increase a lot for low hydrate saturations indicating a significant stiffening of the sediment which – according to effective medium models – would only be achieved by cementing the grain contacts.

*Authors: For the experiments with model sediments we used sand compacted only by hand. Therefore at the initial state the sedimentary matrix should also have nearly no stiffness. The tomography images show that gas hydrate (GH) crystals in our system grow preferentially at narrow throats that should restrict grain movement with water cushion between quarts and GH. The stiffness of the sample will increase with a progressing restriction of movement of quartz grains.*

Comparing your numerical modeling results to effective medium models (Dvorkin, Helgerud, Ecker) and laboratory data (by Priest, Kneafsey, Waite, etc.) might be a good benchmark for your numerical modelling results. This section could benefit from a more thorough discussion about factors causing differences between modelled and laboratory / field data.

*Authors: First of all, we need to point out that this paper mainly focuses on the method of how to prepare synchrotron/CT data for further modelling. While synthetic, simplified models are easier to work with, natural sedimentary matrices require far more careful, ideally, artifact free, approach and the choice of the correct processing is by far not trivial. Some glimpse on the complexity of real systems has been already presented in Chaouachi et al. 2014. Here, we place special attention to the influence of various segmentation techniques to the faithfulness of the reconstruction. Our paper demonstrates this issue and shows possible ways to handle such*

*matter. The modeling itself, using various synthetic or real sedimentary matrices (from tomography), is not the main subject of this paper and indeed would require far deeper discussion.*

The authors conclude that this study enables to distinguish gas hydrate from a gas enriched system and gas hydrate from a free gas system based on their seismic response. However, the modelled velocities for both differ by just 40 m/s. That would actually indicate that both formation mechanisms lead to similar elastic properties. Observed differences from the pore-scale imaging of these two gas hydrate types are not reported in the study.

*Authors: We know the formation process and later we model - not the other way round. The experiments were carried out using different formation processes. For example, the experiment starts in free-gas environment where gas bubbles float freely in the sample. The reaction starts at the gas-water interface, the formation process proceeds until full-transformation state is reached. Then the pressure is decreased, still the same sample, and the hydrate dissociates which means a gas-enriched water is present and the formation process is started again from the gas-enriched environment. That means we know from the formation scenario when a free-gas or a gas-enriched environment is present. The velocities are modelled for both end-members free-gas and gas-enriched environment. Indeed, both formation mechanisms lead initially different distributions of gas hydrates but tend towards similar microstructures – as spelled out in the G-cubed publication by Chaouachi et al. (2015). Yet a convergence to identical structures is not achieved in our samples thus a small difference in seismic velocities are obtained.*

The study is certainly worth publishing but could be improved by relating modelling results to published velocity data.

*Authors: As we explain earlier the actual modeling using various formation scenarios (not the example presented in the manuscript) and deeper discussion of other laboratory and field studies is not intended for this manuscript.*

**Page 2**

**Line 20**: "relies" instead of "relays"
*Authors: Corrected.*

**Line 22**: "has been interpreted before" instead of "has been earlier interpreted"
*Authors: Corrected.*

**Line 27**: I'm not quite sure what you mean by "habits" – it also occurs in the figure caption of
*Authors:* **Habits** *– scientific terminology in the field of geology/mineralogy describing a mineral's appearance. We find this term necessary to be mentioned as we investigated on hydrate crystals.*

**Figure 1**. Can you maybe rephrase it? E.g. hydrate distribution
*Authors: Changed to 'hydrate distribution'*

**Page 3**

**Line 4**: "systems" instead of system
*Authors: Corrected.*

**Line 12**: "these" instead of "this" and "met" instead of "meet"
*Authors: Corrected.*

**Line 14**: I'm not sure what you mean by "refractory" here (that may be my lacking knowledge of the terminology though)

*Authors: An interaction of the sample material with the X-ray beam takes always a form of a complex refractive index with two components being absorption index (amplitude reduction) and refractive index decrement (phase-shift). Depending on the sample-detector distance one component will dominate over the other one; absorption at d=0 and phase shift at d>>0. The size of our environmental cell imposed a particular geometry of the setup where a distance between the sample and detector had to be substantial. Consequently both components contribute to the projections of the refractive index distributions. On our images the absorption is mainly responsible for the global grey levels in the sample. The refractive index decrement sands mainly behind a strong edge enhancement, the improved contrast at transitions between mediums with different electron densities. Measurements in such mode in late 90s have been dubbed Refraction-enhanced x-ray imaging".*

**Line 24**: instead of "followed" you could use "monitored" or "observed"
*Authors: Changed to "monitored"*

**Page 4**

You're mentioning the density differences between water and methane hydrate vs. Xenon hydrate. I think it would be helpful to include a table with densities for your sample components (grains, Xe hydrate, Xe gas, water)
*Authors: Since absorption of X-ray beam energy is not only depending on the the material's density but also: a) thickness of the material, b) wave length [lambda] and c) the ordinal/atomic number Z we only listed the densities of the hydrates in Table 1. We find that the information on density is most important in the last paragraph.*

**Page 5**

In the first paragraph it sounds like you're saying that Xenon gas and water could not be distinguished (and it sounds like that again on Page 8, line 14) yet in the second paragraph (line 11) you say you were able to distinguish between gas and water. Can you clarify this?
*Authors: We slightly rephrased the paragraph to make it clearer that we are able to distinguish the phases hydrate, water and gas-filled pores by eye using the tomograms BUT for the segmentation (binarization) of the images we faced major difficulties. This process was more than tricky and is described in the following workflow.*

*Rephrased text passage (p.4, line 27-29): "The reconstructed tomograms revealed that in spite of contrast enhancement with Xe gas the grey value differences between the gas and water phase were distinguishable by eye but too low for further segmentation approaches"*

**Line 11**: "commonly occurred" instead of "occurred most"
*Authors: Corrected*

**Line 26**: "prior to" instead of "prior"
*Authors: Corrected*

**Page 6:**

**Line 14**: "number of iterations" instead of "number of iteration"
*Authors: Corrected*

**Page 8:**

**Line 3**: "meets" instead of "meet" and a comma comma after "meets"
*Authors: "Meets" was corrected*

**Line 11**: "then" instead of "than"
*Authors: Corrected*

**Line 14**: "water or gas-filled" instead of "water of gas-filled"
*Authors: No, we meant the water within the gas-filled pores.*

**Page 9:**

**Line 2**: just "yielded" instead of "yielded in"
*Authors: Corrected*

**Line 23** "obtained" instead of "sustained"
*Authors: Corrected*

**Paragraph 2**: I think it would be helpful if you mentioned the laboratory derived values by for porosity and permeability by Madonna et al. to compare to your numerically derived values

*Authors: The information on the lab data has been added (p.9, line 25-27): "Digital rock analysis indicated a total connected porosity of 18% and a permeability of 150 mD for Ber1b where the lab results revealed a connected porosity of ~ 20%. The Permeability for this sample as provided by the Berea Sandstone$^{TM}$ Petroleum Cores (Ohio, USA) is between 200 and 500 mD."*

**Page 10:**

**Line 21**:"multiple" instead of "multiply"

*Authors: Corrected*

**Line 24**: Can you elaborate a bit more of what you mean by "benchmarked" here? Did they compare modelled and laboratory measured data?

*Authors: Indeed, computing effective properties the authors of Andrä et al. compared modelled and lab measured data. The work has been presented in two papers: "Digital rock physics benchmarks – Part I: Imaging and segmentation" and the complementing second publication: "Digital rock physics benchmarks – Part II: Computing effective properties" . For further details please see:*

- *Andrä et al. (2013a): "Digital rock physics benchmarks—Part I: Imaging and segmentation."                                                                         DOI: http://dx.doi.org/10.1016/j.cageo.2012.09.005*
- *Andrä et al. (2013b): "Digital rock physics benchmarks—part II: Computing effective properties." DOI: http://dx.doi.org/10.1016/j.cageo.2012.09.008*

**Line 31/32**: voxel sizes are missing units
*Authors: Lines 33/34 have been rephrased and corrected. New sentence: "cropped dataset of 1800 x 1800 x 1800 voxels each of $(0.74 \ \mu m)^3$ in size was reduced to a cube of 400 x 400 x 400 voxels each  of $(3.4 \ \mu m)^3$ resampled size."*

**Page 11/12**

Note: most of the things I mentioned in my short comment are referring to these pages. What makes me wonder most is that your increase in velocities is really high for a hydrate saturation of 17% (for both tested formation mechanisms). So would you conclude that the hydrate significantly stiffens the sediment even though the hydrate does not appear to actually touch the grains?

*Authors: Yes, exactly this has been stated in our first paper (Chaouachi et al. 2015). The fluid film is very thin – we did not observe the grains touching the hydrate even though the images are restricted to resolution limitations. This fluid film could be one explanation for squirt flow mechanisms leading to anomalous seismic responses which is currently under numerical examination from our side and will be hopefully published at the end of the year.*

You could add some more detail here to your discussion. I think this is actually really interesting! Your images show that the hydrate doesn't follow any of the end-member models discussed by effective medium theory (pore filling, contact cementing etc.).

*Authors: A pure pore-filling and frame/matrix-supporting distribution types are not found in our samples. Instead we clearly observe microstructures that coat quartz grains and fill pore spaces. This remarkable mixture of various configurations is additionally complicated by the fluid film causing the grains not to touch the hydrate. Consequently a purist grain supporting model is likely to be never observed for gas hydrates in quartz sediments! It is more like "an egg*

*in an eggcup" scenario. We also do not observe a pure form of a pore filling model. Yet, please note that we did not exhaust all possible scenarios and this particular microstructure might exist under full water saturation as suggested by Kerkar et al. 2014, Geochem. Geophys. Geosyst., 15, 4759–4768, doi:10.1002/2014GC005373*

So maybe one of your conclusions could also be that we need better physical models than the effective medium ones.

*Authors: The established physical models of how hydrate is distributed in sedimentary matrices are a very good attempt to simplify the complex system. It is nevertheless already clear that nature does not follow simple paths and indeed mixed models are needed for more accurate evaluations of gas hydrate saturations. The observed thin fluid layer needs to be further investigated and the physical models improved.*

It would also be interesting to discuss whether your images showed any differences in hydrate distribution for the two formation mechanisms (from free gas + water and from gas enriched water). The literature usually assumes the first one forms cementing hydrate while the latter forms pore filling hydrate. It seems like you didn't observe this difference – neither in your images nor your modelled velocities. That's an interesting observation and worth discussing!

*Authors: This has been discussed in the earlier paper and will be again in detail in the planned publication. On the side note, fresh microstructures may show some differences due to somewhat different packing and crystal size of gas hydrates (that in turn influence the distribution and local thickness of the water film) but in course of time due to the crystal coarsening both scenarios are likely to provide a similar response to the propagation of acoustic waves.*

**Line 17 to 21**: If I'm not mistaken the conclusion from the studies your mentioning (Waite, Priest etc.) is usually that the velocity is higher for hydrate formed from free gas and water than for hydrate formed from gas-enriched water (especially at low saturations, like the ones you used for your experiments). Your model indicates something different. You could add some phrases to hypothesize why your model results differ from lab data.

*Authors: There might be several reasons for this contrasting observation. One could be the missing pressure-induced stress (uniaxial/triaxial) during measuring. Another the way the samples were prepared (unconsolidated sand instead of pre-compacted sand), full vs. partial water saturation, variable water content and its distribution in partially saturated systems, etc. We would like to emphasize again the fact that the model results are still under examination and were in the article to show what we can do next with our imaging results and data processing. The numerical results are in a reasonable range which was important for us. Now, we agree further modelling needs to be done to establish a conclusion on this.*

**Anonymous Referee #3**

This manuscript details the processing of x-ray scans of hydrate-bearing sands to derive high-resolution 3-D CT representations of the pore-scale geometry, which can then subsequently be used in numerical models to better characterize the geophysical properties of these materials. This manuscript has the potential to be of great value in terms of improving 3-D imaging of rock samples as well as improving our understanding of hydrate-soil interaction at the pore-scale. However, there are a number of clarifications that I feel need to be addressed to ensure the significance of this paper. The main clarification I feel is required is related to Section 3. Further details are required to describe the underlying constitutive model that is considered in the numerical model. As noted by the other referee there are a number of rock physics models that have been developed to describe the interaction of the hydrate and soil grains, which give rise to significant differences in wave velocity at the reasonably low hydrate saturations used in your modeling. The wave propagation would depend not only on the respective elastic moduli of the components in the system but how these components interact. Details of the modeling, as in size of model, mesh size, how the discrete nature of particles and hydrate are modeled is lacking. These issues need to be addressed for this aspect of the paper to be worthwhile.

*Authors: Referring to the concern that the details of modelling are missing: All noteworthy details are given in the relevant sections (grid size = 400x400x400). The discrete nature of the particles and hydrate is in form of segmented data the direct model input.*

In addition there appears to be some ambiguity to exactly what phases are present in the samples tested.

*Authors: In Figure 4 all phases (Quartz, hydrate, water and gas) appearing in the tested samples are depicted. There is no ambiguity to it.*

Were both systems 'excess gas'? or are you inferring the 'gas enriched' is 'excess water'?

*Authors: Pore spaces investigated in our primary environment were only partially occupied by juvenile water. The remaining pore volume filled gas under pressure. Here hydrates are formed at the gas-water interface. In the second scenario pore spaces are partially filled with water enriched in gas (trough melting of gas hydrates) that is in a contact with free gas under pressure. Concluding, in both scenarios pore spaces are partially occupied by gas under pressure (excess gas condition). In the second case, hydrates are reformed from gas enriched melt water; a decomposition product of hydrate crystals grown in the first formation scenario.*

Given the low velocity contrast between the two methods are they forming hydrate at the same location (relative to water) and therefore the minor variations in velocity are related to just reforming characteristics.

*Authors: The location of gas hydrate crystals during the formation and reformation can be initially very different on a local scale. Also microstructures are not the same. Yet, as the reaction closes to completion (free water largely transformed to clathrate) these differences diminish. It*

*should be noted that the crystal size at this stage is still somewhat different (coarsening is much slower than the crystallization) that in turn will affect the thickness and distribution of the water film between quartz and gas hydrate crystals. This may be also the reason for small differences in the velocities. It is still an unproven hypothesis but will be subject of further investigations.*

A final ambiguity I feel that needs to be addressed is the link between the title of this manuscript and the overall thrust of the paper. The title suggest that the focus of the paper in on the processing of the synchrotron data, however the conclusions focus nearly entirely on the success of the numerical modeling in determining seismic response.

*Authors: We have reworked the conclusions to better match the title.*

I have attached an annotated manuscript highlighting some suggested grammatical corrections and areas where the grammar should be improved/revised.

Please also note the supplement to this comment:
http://www.solid-earth-discuss.net/se-2016-54/se-2016-54-RC3-supplement.pdf

**Page 1**

**Line 11**: *"To date, very little is known about the distribution of gas hydrates"* I think we know quite a bit about the distribution so much about the actual interaction

*Authors: Unaltered observations of gas hydrate in natural sediments, in particular of low-concentrations, on the sub-microscale have not been reported so far. Larger gas hydrate pieces that have a chance to survive the recovery also carry signs of alterations. Therefore original microstructures and fabric (distribution in pore spaces) are actually still largely unknown. As the reviewer correctly noticed, even less is known about the nature of contact between gas hydrates and various minerals that can be pivotal for e.g. the propagation of acoustic waves. The progress in pressure coring over the last years gives some hope for intact samples but working solutions developed at e.g. National Energy Technology Laboratory (NETL) are still not available to the broader community.*

**Line 12**: *"affecting the seismic properties at low hydrate concentration"* identify the 'what'....matrices, rock

*Authors: The reviewer rightfully noticed an inconsistency in the text. We have modified the fragment in question to rectify the mistake: "To date, very little is known about the distribution of gas hydrates in sedimentary matrices and its influence on the seismic properties of the host rock, in particular at low hydrate concentration."*

**Line 18**: I have no idea what you mean by *'model-free deduction'*? This sentence needs to be reworded to make it explicitly clear what you are trying to say

*Authors: Indeed the sentence was somewhat awkwardly written. We have replace it with the following one: "A combination of the tomography and 3D modelling opens a path to a more reliable deduction of properties of gas hydrate bearing sediments without a reliance on idealised and frequently imprecise models."*

**Page 2**

**Line 15**: this is two environments
*Authors: The sentence has been corrected to "Two of those difficult environments…"*

**Line 17**: in what sense are the surveys intensive?

*Authors: Natural deposits are known for many decades but the real interest in these solids came after the unquestionable success of Mallik exploratory well (1998-2008). Natural gas hydrates were placed in a spotlight as a potential future source of hydrocarbons. In spite of this attention there was a substantial lack of precise estimates of potential deposits and recoverability; Estimates that at that time were scattered even by a few orders of magnitude. Even less was known about environmental and geotechnical aspects of deposits. This is the place where numerous exploratory and monitoring ship/land surveys came into play; some of them focused only on the acoustic and/or EM investigations, other also involved gravity or pressure coring, well logging, long term observatories etc. In course of time it became also clear that not all deposits are optimal for the exploitation and the ones that could be interesting are often discontinuous forcing research groups to go for data heavy, higher resolution acoustic and EM techniques, denser profiling grids and more frequent coring. All these activities were led by countries like Japan or USA but also e.g. Germany, South Korea, China, India, New Zealand, Norway or Brazil was very active in this area. Reports form these studies systematically resurface in large quantities in form of scientific papers, conference proceedings or cruse and well test reports; too many to list here. Revives of these activities are summarized every several years by e.g. Tim Collet from USGS and his colleagues (e.g. Collet et al. 2015, Methane Hydrates in Nature-Current Knowledge and Challenges, JOURNAL OF CHEMICAL AND ENGINEERING DATA Volume: 60  Issue: 2  Pages: 319-329 DOI: 10.1021/je500604h.)*

**Page 3**

**Line 8:** model or images?

*Authors: We find "models" more meaningful than" images". Please note that the word in question should be red in the context of the whole sentence. Here we clearly refer to "traditional models" that are microstructural concepts and not images.*

**Line 10:** I guess you are specifically referring to attenuation, since difference in velocity may be minor?

*Authors: Exactly, this paper discusses the prior steps to achieve modelling of attenuation on a sub-micron scale using synthesized samples of hydrate bearing sediments as a direct model input.*

**Page 3**

**Line 17:** Data acquisition - is this an adequate section title for this section? doesn't really identify the fact that this section is dealing with experiment set-up etc.

*Authors: With the current structure it might be indeed not the most complete title. Therefore we have updated the section title to: "* Experimental setup and data acquisition*".*

**Line 24:** is 'the hydrate formation process followed' in the custom or is it 'hydrate formation carried out'

*Authors: "followed" has been substituted by "observed" which was also suggested by Reviewer #2.*

**Page 4**

**Line 2:** is this true? I would not believe that the sample holder was chosen to match the numerical model, but almost entirely chosen due to x-ray imaging constraints?

*Authors: Actually it is true. During the design phase we have carefully matched restrictions and requirements of both techniques. The modelling required primarily a sufficient the model grid size, i.e. number of grains in the reconstructed volume. This in turn increased the absorption from the sand-hydrate matrix and effectively forced us to trim Al-walls from the sample holder, volume of free Xe-gas and walls of the polymer dome.*

**Line 5: ...**which would hinder data processing...

*Authors: Correct, therefore methane was replaced by xenon.*

**Line 9:** why exactly is this step carried out (Author's note: referring to compression & decompression cycles)

*Authors:* A sample insertion is performed under ambient atmosphere. After closing the cell the experimental volume is still filled with air that is purged through a compression with Xe gas and subsequent depressurization. Through several compression-decompression cycles air components are diluted to negligible concentrations.

**Line 14:** what does 'objectives' mean in this context?

*Authors: At the TOMCAT beamline synchrotron light is converted to a visible one on the LuAG:Ce scintillator foil that is further on guided though a selected microscopic objective to achieve a desired magnification. We have somewhat modified the sequence of sentences to better emphasize this fact.*

**Line 27:** already stated above

*Authors: No, this was not stated so far.*

**Line 28:** should state that 'gray scale difference' is related to density

*Authors: The word density would imply that we hold information on the relation of grey values to electron densities in a voxel. This is nevertheless not the case as measured values are actually a convolution of the absorption and phase shift. Without a proper phase retrieval we can talk only about relative differences, e.g. Xe-hydrate appears brighter than Quartz even though Quartz has a higher density than Xe-hydrate.*

**Page 5:**

**Line 1:** what does this mean? did you do a new test?

*Authors: The original sinograms have been reconstructed using another algorithm than the regridding algorithm (Stampanoni & Marone). The aim was to enhance the contrast of the gas/water phase to be able to segment/digitize those phases.*

**Line 9:** this section is called 'image enhancement' but the text on this page has nothing to do with image enhancement?

*Authors: Agreed, we changed the Caption, so the following paragraphs on image enhancement refer to the heading which might be clearer to any future reader.*

Also lots of words just to say you 'sub-sampled'?

*Authors: Subsampling was accomplished in several cases: to divide the dataset into smaller portions applying the image enhancement techniques than merge the parted dataset together again. This was done due to the rather big data size (24GB) and therefore high computational demands.*

**Line 25:** doesn't make sense, not sure what you are saying?

*Authors: In CT imaging (computed X-Ray tomography) we produce 3D images of real existing materials - a stack of numerous 2D images produce a 3D image that is why the 3D images are based on 3 spatial directions which are XY-plane, XZ-Plane and YZ-plane. For image artifact evaluation and selection of filters it is necessary to rate the slices but not only in one direction but all three.*

**Page 6:**

**Line 6a:** Reviewer wants to delete parts of the text "full-transformation state"

*Authors: The phrase 'Full-transformation state' cannot be deleted as it is a mandatory information since we could have chosen datasets where the full transformed state was not achieved (e.g. during the formation process right after nucleation took place). We have nevertheless somewhat modified the sentence for better readability: "Five datasets which*

*derived from samples containing approximately 17 Vol% of hydrate (free pore water is fully transformation to hydrate) were selected."*

**Line 6b:** not sure why this is here? relevance?

*Authors: ~17 Vol% of hydrate is in the full-transformation samples - we needed to validate this amount in the DRP numeric simulation when estimating the hydrate amount from the images after image enhancement which differs a lot depending on the filter type.*

**Page 8:**

**Line 3:** "water" not quite sure what this means? need to clarify for the typical reader who may not be proficient in segmentation techniques

*Authors: The "water" is used here as a metaphor, describing the technique of watershed segmentation. The easiest way to understand the watershed transform is to think about two valleys separated by an elongated ridge. Water poured from above falls into one valley or the other one with the ridge shedding the stream into two. This naturalistic picture can be further generalised to a surface with two depressions (catchment basins) separated by a watershed line, the ridge, as on the figure below. The watershed segmentation can find the catchment basins and watershed lines for any grayscale image.*

[Figure]

*(Source: http://www.mathworks.com/cmsimages/65312_wm_watershed_fig9_w.gif)*

*The watershed segmentation is also coupled to region growing algorithms where the "water" propagates until the watershed line is found.*

**Line 24:** The reviewer wants to delete "... the material under examination..."

*Authors: Suggested change would lead to a generalization of the sentence. Yet, the text refers to a very specific material and therefore we are reluctant to agree with the reviewer.*

**Line 27:** not clear what 'such study' you are referring to. Also have you stated that the above image enhancement does include 'side effect'.

*Authors: This is true "side effects" have been mentioned but with the help of this study we are able to approximate the effect of the image enhancement on the modelling result. Also there is a need to show this since there is a significant trade-off in the results depending on the image enhancement technique*

**Line 30:** I am a little bit confused. Have you carried out the pre-study or was it carried out by Saenger & Sell.

*Authors: This pre-study was presented as a poster at the EAGE 75th conference in June 2013 (London, UK)*

**Page 9:**

**Line 11:** what further numerical investigations were done? How? What methods

*Authors: Numerical investigations to model porosity and permeability were carried out using PoroDict and a Boltzmann solver of the software package GeoDict (Math2Market). For the effective p-wave velocities the model of E.-H. Saenger was used. Note: PoroDict was used also to estimate the Hydrate content in the samples described her.*

**Line 17:** ref?

*Authors: This is a modelled result obtained by the method mentioned.*

**Page 10:**

**Line 11:** how big is your FD model? what is the grid size? does it pick up individual phases?

*Authors: Since the original segmented/digitized 3D image would be too large to serve as a direct model input. Therefore it was decided to use a resampling tool to decrease the number of grids. After careful testing we chose the 'Lanczos method' for resampling to gain a 400x400x400 grid (= amount of voxels). Those pick up individual phases which have been segmented prior.*

**Line 31:** is the micrometer

*Authors: Text passage has been changed to: "….a cropped dataset of 1800 x 1800 x 1800 voxels each of $(0.74\ \mu m)^3$ in size was reduced to a cube of 400 x 400 x 400 voxels each of $(3.4\ \mu m)^3$ in size."*

**Page 11:**

**Line 8:** are the pores filled with water? or Xe gas? not clear

*Authors: The pores are assigned to be filled with water and gas for modelling as we described earlier it was difficult to distinguish between gas and water during segmentation there might be uncertainties left.*

**Line 14:** this needs a ref? who has measured water/gas mixtures under no confining stress. Was your porosity relevant to such tests, since porosity plays a huge part on wave velocity.

*Authors: No one did that this is true and will be one of our discussion points for the next planned paper which will only focus on the modelling part. The main idea of the present paper is to show a workflow to model with experimentally derived 3D images at different hydrate formation stages.*

**Line 19:** i am not sure Priest et al undertook 'gas rich' or gas-enriched' tests

*Authors: The "gas rich" is indeed more accurate. The paper in question indeed discusses experiments with a similar philosophy behind. Also the pore spaces were only partially filled with free gas. The major difference is found at the beginning of the formation where the nucleation is initiated on refrozen and not liquid water. Moreover, this earlier work covers also a much higher range of saturations.*

**Line 21:** Carcione and Gei measured hydrate saturation from Malik. The stress conditions, hydrate saturations etc were markedly different from your test? Was the hydrate in nature formed in a  way similar to yours. Reidel and Yuan are based on deep ocean, so again entirely different stress conditions to that at Malik. So would the the impact be on your modeling? So what range of P-wave velocities were observed?

*Authors: Obviously stress fields may improve grain-grain contacts. How the hydrates at the Mallik site were formed is open to speculations – we do not want to enter this discussion with so little information at hand. Still, we think that, both in nature and laboratory experiments, the same fundamental interactions are operating. Yet, as mentioned above, different laboratory experiments may not have converged to a stable picture for a given hydrate saturation at the moment when experimental data were taken.*

**Line 22:** slightly lower velocity but significantly lower hydrate? how do these two contrasting aspects explain things?

*Authors: Yet, we are not sure as we mainly focused on the relative P-wave velocity at different hydrate saturations. Since the upscaling from microscale to field-scale is a true challenge we are not able to calibrate our numerical results derived on the microscale. This paper is a method paper representing the first results on the modelling. The next step is further modelling approaches to obtain Vp, Vs and attenuation linked to the data and which will hopefully clarify this, including the aspects of preparation-dependent differences of microstructures discussed above. We updated the text passage to the following:*

*Page 11: (now) Line 24 - 29 "With the presented results of the modeled p-wave velocities the model approach is in a realistic range when compared with field (Carcione and Gei, 2004;Riedel et al., 2002;Yuan et al., 1996) and laboratory data (Zhang et al., 2011;Priest et al., 2009;Priest et al., 2005); but it is noteworthy, that the modelled results give only slightly lower values than the experimental ones,  even though at significantly lower hydrate saturation (<20%) compared to*

*results from laboratory work. Furthermore, intensive modelling on various scans needs to be performed to fully understand the apparently contrary observation."*

**Line 25:** that is true, but you are numerically modeling the system. You can choose what you want?

*Authors: No, we do not have this flexibility. The model takes directly segmented 3D images as input. Therefore, the matrix and hydrate distribution are given solely by the experiment.*

**Line 29:** the general conclusions do not really match the thrust of the paper, nor do they align with the title? Need to frame the conclusions to the thrust of the paper and the title

*Authors: Yes it does need a slight rephrasing. The general thrust of the paper is to show a workflow to gain further knowledge from synchrotron-based tomography on gas hydrate bearing sediments and the needed data-processing steps. The Vp modelling presented in the last paragraph is just an example of what we can do in the first glance with the resulting image data - that is why the title includes "on the path". The next paper on the scope of Vp/Vs/attentuation modelling with this data is planned.*

**Line 30:** poor sentence. Difficult to understand exactly what you are trying to say

*Authors: What we wanted to point out is that in DRP (Digital Rock Physics) it is necessary to conduct preliminary studies concerning image enhancement especially if there is no access to lab results.*

**Page 13:**

**Line 27:** pages before year?

*Authors: Corrected.*

**Page 16:**

**Line 10:** Why capitalise?

*Authors: Corrected.*

**Line 14:** no year?

*Authors: The publication date (2011) has been added.*

**Page 17:**

**Line 26: gap**

*Authors: Corrected*

**Page 20:**

**Figure 1:** is it not normally referred to as frame supporting?

*Authors: Both expressions are widely used in literature which are literally the same (sediment-) frame supporting = (sedimentary) matrix-supporting*

**Line 4: gap**

*Authors: Corrected*

**Page 24:**

**Line 4**: successful? since it is obvious that some form of segmentation has occurred

*Authors: No, only filters where a porosity value is given were successfully segmented. The images you see in the figure do not represent segmented data but only filtered. Segmentation means to reclassify the image in order to get rid of single grey values and gain segmentation classes (e.g. phases) as a result.*

**Page 26:**

**Figure 7**: what exactly is the problem? is it that the green colour is observable? what do the different colours mean?

*Authors: Please compare both images carefully. You will notice significant errors especially when focusing on the white circled areas. Black spots are equal to blank non-segmented areas where the segmentation attempts failed due to image quality issues. The colors represent binarized/digitized phases whereas significant problems occurred. Note: The colors are randomly picked and represent the segmented phases.*

**Page 27:**

**Figure 8:** on the right (author's note: refers to the segmented dataset) & is the pore space gas or water? mixture?

*Authors: The caption ''segmented dataset'' is on the figure but we added 'on the right' for better understanding. In this example the pore space can be assigned as both: water or gas. Here one sees a successfully segmented dataset (as labeled in the image) compared to the initial filtered image this is fundamental for segmentation processing and represents the desired result – an image with assigned uint 8bit labels (0,1,2,3) instead of grey values ranging from 0-65478*

**Page 28:**

**Figure 9**: not sure what this image is saying? image on right not very clear?

*Authors: This image shows the triangulation of the grain's surface since this is also a very important information concerning numerical simulations in 3D it is depicted here.*

**Page 29:**

**Figure 10:** why is there less hydrate in 10.3 compared to 10.2? Is the centre figure and fig 5 necessary. Would be good to know formation step time? hydrate saturation etc for 10.2, 10.3 and 10.4

*Authors: Good point! 10.3 and 10.2 were mixed up and has been corrected. Detailed information on the formation process in given in Chaouachi et al. 2015: DOI 10.1002/2015gc005811*

**Page 30:**

**Figure 11:** not sure this image tells us anything different from fig 7? certainly harder to visualise

*Authors: In Figure 11, the thin water film in-between the hydrate and grains is depicted – this is the result of all data-preprocessing, segmentation and rendering. Figure 7 has nothing to do with a rendered image – as already explained Fig. 7 shows segmentation errors during the binarizing process.*

**Page 30:**

**Figure 13**: what does the min max scale mean? Vp? Hydrate? put some scale on it?

*Authors: The scale refers to Vp which has been added now.*

**Page 33:**

**Figure 14:** state what x axis is. Is the y-axis in the usual form for given and axis title?

*Authors:  As mentioned in the figure caption the results for a time series of scans is shown. 1 = first scan, 2 = second scan and so on.*